# PLOT: Pseudo-Labeling via Object Tracking for Monocular 3D Object Detection

## Abstract

Monocular 3D object detection (M3OD) is crucial for scalable perception across fields like autonomous driving, robotics, and surveillance. However, progress is hindered by limited 3D annotations and the inherent ambiguity of single-image geometry. Current methods often rely on strong geometric assumptions or carefully curated datasets, which limit their applicability to real-world scenarios. In this paper, we present **PLOT** (**P**seudo-**L**abeling via **O**bject **T**racking), a framework that generates 3D annotations from monocular videos without auxiliary sensors or model retraining. PLOT tracks object and background trajectories to estimate camera motion and perform object association in pose-unknown settings. These trajectories are integrated through the shape fusion of frame-wise pseudo-LiDARs, yielding reliable annotations under occlusion and viewpoint shifts. Recognizing temporal coherence as a fundamental requirement for reliable shape fusion and video perception, we design a global object memory that preserves consistent object identities across frames. PLOT achieves robust annotation quality and strong generalization on both M3OD video benchmarks and in-the-wild videos, proving its effectiveness across diverse and unconstrained domains. The code and weights will be publicly released upon acceptance.

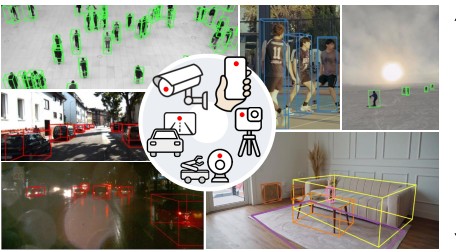 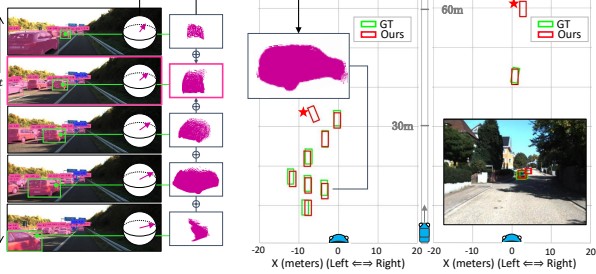

(a) Labeling results across diverse domains      (b) Multi-frame aggregation and BEV comparison with ground-truth

Figure 1: **PLOT** (**P**seudo-**L**abeling via **O**bject **T**racking) generates accurate 3D labels directly from monocular videos without requiring auxiliary sensors or training, as illustrated in (a) qualitative results across diverse scenarios. Furthermore, (b) our object tracking and aggregation pipeline produces shape-complete pseudo-LiDARs, yielding BEV maps comparable to ground truth—again without any training-and can identify additional objects (marked with a red star).

## 1 Introduction

Monocular 3D object detection (M3OD) has emerged as a critical perception task for vision-based autonomous systems, enabling applications across autonomous driving, robotics, surveillance, and retail, offering a cost-effective alternative to LiDAR-based methods (Shi et al., 2019; Hu et al., 2022; Pan et al., 2024).While recent years have witnessed significant advances in M3OD (Chen et al., 2016; You et al., 2019; Reading et al., 2021; Zhang et al., 2023; Jiang et al., 2024b), its real-world deployment remains limited by two key challenges: the limited availability of high-quality 3D annotations and the inherent ambiguity in lifting 2D images to 3D.

As a result, progress in M3OD has been largely confined to curated benchmarks in sensor-rich environments—such as autonomous driving (Geiger et al., 2012; Sun et al., 2020; Caesar et al., 2020) or indoor scenes (Dai et al., 2017; Baruch et al., 2021). This has limited the generalization of M3OD models beyond these curated scenarios. Although recent efforts (Brazil et al., 2023; Li et al., 2024; Zhang et al., 2025a) have attempted cross-domain training by combining multiple datasets, they often fail to generalize to unconstrained environments with moving cameras, unknown poses, and diverse viewpoints, as illustrated by their poor zero-shot performance on in-the-wild data (Milan et al., 2016; Contributors, 2025) (see Fig. 2). While weakly supervised methods have been proposed to alleviate the need for dense 3D annotations, their dependence on sophisticated priors (Jiang et al., 2024a) and auxiliary data (Peng et al., 2022b; Tao et al., 2023) still hinders generalization.

Recently, pseudo-labeling has been explored to generate 3D annotations without explicit 3D supervision (Zhang et al., 2024b; Huang et al., 2024b; Liu et al., 2024). However, most existing methods operate on single images, limiting their ability to resolve occlusions and to disambiguate object orientation and scale. These limitations result in noisy annotations that propagate errors during both training and evaluation. In addition, many of these pipelines rely on sensor poses (Skvrna & Neumann, 2025) and hand-crafted geometric heuristics (Huang et al., 2024b), restricting their scalability and adaptability across domains.

Figure 2: **Zero-shot estimation on MOT17** (Milan et al., 2016) **and Pexels** (Contributors, 2025)**:** Predictions from (left) an open-set M3OD model (Yang et al., 2025) and (right) our pseudo-labeling-based method. Our method performs well on out-of-domain camera views (zoom in for a better view).

In this paper, we propose **PLOT** (**P**seudo-**L**abeling via **O**bject **T**racking), a framework that generates reliable 3D annotations from any monocular video without requiring auxiliary sensors or model retraining. Video streams are abundant and encode rich geometric information over time, yet using them requires solving key challenges in object association, label consistency, and camera motion. Recent 3D reconstruction methods (Leroy et al., 2024; Wang et al., 2025a; Zhang et al., 2025b; Wang et al., 2025b) have shown remarkable progress in recovering global scene structure from unstructured image collections or videos. However, these approaches remain primarily designed for static environments: they either ignore dynamic objects altogether (Leroy et al., 2024; Wang et al., 2025a) or handle them only in constrained forms (Zhang et al., 2025b; Wang et al., 2025b). As a result, they are not well suited for object-centric 3D perception, where the objective is to estimate the geometry, pose, and motion of individual objects rather than reconstruct a single static background scene. Building on the established tracking model (Harley et al., 2025), we focus on the downstream challenge of associating object observations over time—a critical yet underexplored step that we explicitly formalize and address.

Specifically, PLOT builds upon off-the-shelf 2D detectors (Ren et al., 2024) and monocular depth estimators (Piccinelli et al., 2024; Wang et al., 2025c) to obtain frame-wise observations, which are subsequently aligned using dense point tracking (Harley et al., 2025). This enables: 1) trajectory-guided shape fusion, producing unified pseudo-LiDAR representations, and 2) estimation of object and camera dynamics for robust pseudo-labels. To reliably handle objects reappearance and exit events-and to further improve annotation quality in cluttered scenes with imperfect detections-we introduce a Global Object Memory (GOM) that enforces video-consistent label association and mitigates identity switches across frames. This module improves robustness under real-world occlusion, detector noise, and dynamic backgrounds.

While our framework is instantiated for monocular 3D object detection, it addresses broader challenges in video-based 3D understanding—most notably, how to reliably associate object instances without sensor pose or calibration data. Through dense point tracking and GOM, our method provides a practical blueprint for learning-free association in pose-unknown settings. This also expands the applicability of M3OD to sensor-infeasible or unstructured environments, helping to democratize 3D annotation in the wild. PLOT is validated on various M3OD video benchmarks (Geiger et al., 2012; Liao et al., 2022; Sun et al., 2020) and unconfined monocular videos (Milan et al., 2016; Ding et al., 2023; Contributors, 2025), demonstrating strong performance in various scenarios.

## 2 RELATED WORK

**Supervised monocular 3D object detection (M3OD).** Early supervised M3OD methods (Brazil & Liu, 2019; Wang et al., 2021; Peng et al., 2022a; Lu et al., 2021; Jia et al., 2023) achieve progress in curated settings but remain restricted by scarce 3D annotations. To alleviate this, several approaches use external data such as video (Brazil et al., 2020; Wang et al., 2022), LiDAR (Huang et al., 2022), or CAD models (Liu et al., 2021; Parihar et al., 2025). More recent works move toward open-set detection, exemplified by Cube R-CNN (Brazil et al., 2023) and successors (Yao et al., 2024; Zhang et al., 2025a; Yang et al., 2025), which exploit large-scale benchmarks (Brazil et al., 2023) and 2D foundation models (Ravi et al., 2024; Oquab et al., 2023). While these broaden domain coverage, they remain tied to 3D ground-truth supervision and still struggle to generalize consistently in unconstrained videos (see Fig. 2).

**Weakly-supervised M3OD.** Given the challenges of cost and scarcity of 3D annotations, several works (Qin et al., 2020; 2021; Han et al., 2024; Jiang et al., 2024a) have pioneered weakly-supervised M3OD. WeakM3D (Peng et al., 2022b) estimates 3D attributes given raw LiDAR points, while WeakMono3D (Tao et al., 2023) and subsequent work (Han et al., 2024) utilize multi-view constraints. SKD-WM3D (Jiang et al., 2024a) proposes a self-teaching pipeline to lift 2D features into 3D space using depth completion, while VGW-3D (Huang et al., 2024a) improves performance with perspective-invariant error prediction and decoupled visual guidance. Importantly, MonoGR-Net (Qin et al., 2021) introduces a general-purpose M3OD model that utilizes video priors. However, these methods require sophisticated priors, which limit their scalability. GGA (Zhang et al., 2024a) addresses this issue by introducing a generalizable weakly-supervised M3OD framework that leverages general geometric priors and 2D ground-truths. VSRD (Liu et al., 2024) proves the efficacy of pseudo-labeling and silhouette rendering in weakly-supervised M3OD.

**Tracking for 3D perception.** Tracking has recently gained attention in video-based 3D perception for its potential to model scene dynamics over time. Seurat (Cho et al., 2025) and ViPE (Huang et al., 2025) utilize dense tracking across frames to enhance 3D perception in dynamic scenes, for the first time. While Seurat focuses on depth map estimation, ViPE integrates tracking signals for broader scene-level understanding. However, neither method addresses object-level geometry or pose. Vid2CAD (Maninis et al., 2022) targets CAD model alignment using multi-frame tracking, but focuses on aligning indoor objects to canonical CAD poses without addressing real-world clutter. In contrast, our approach leverages off-the-shelf dense tracker (Harley et al., 2025) to associate object observations over time and estimate relative camera motion—enabling pose-free, training-free 3D annotation from monocular videos without bundle adjustment or calibration.

**Pseudo-labeling for M3OD.** To address the scarcity of 3D annotations, pseudo-labeling has emerged as a viable alternative for enabling M3OD without explicit 3D supervision. In particular, VSRD (Liu et al., 2024) introduced a multi-view-based auto-labeling approach, showing that reliable pseudo-labels can enhance weakly supervised training. OVM3D-Det (Huang et al., 2024b), meanwhile, proposes a zero-shot labeling framework that utilizes open-vocabulary 2D detectors (Hu et al., 2024; Ren et al., 2024) and LLM priors (Achiam et al., 2023) to estimate 3D box attributes without training. However, it operates on single images and thus remain sensitive to occlusions, noise, and scale ambiguities, limiting their applicability in more complex or dynamic scenes. MonoSOWA (Skvrna & Neumann, 2025), a recent work, similarly leverages per-frame 2D mask association for pseudo-label generation, but the reliance on known camera poses with naive mask-based tracking limits its scalability, and does not address cluttered environments explicitly.

## 3 MOTIVATION

While monocular videos offer a promising avenue for scalable 3D annotation, their direct use without out camera poses introduces challenges underexplored in prior work. We highlight two key problems that motivate our method, with qualitative evidence provided in the supplementary material.

**What are the challenges of object association in a monocular video?** Video-based object association may seem straightforward given recent progress in 2D tracking, but maintaining identity

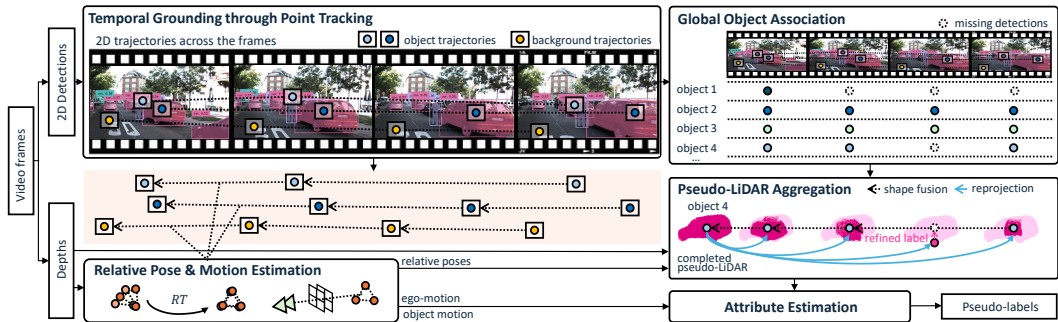

Figure 3: **Overall architecture of PLOT.** Given monocular videos, we extract 2D detections and depth, and track points to obtain temporally grounded correspondences. These are used to estimate relative poses and camera motions across frames, enabling shape fusion and orientation estimation. A global object association module refines trajectories and recovers missing instances. Finally, completed pseudo-LiDARs are reprojected to each frame for consistent 3D attribute annotation.

consistency across frames remains fragile. In realistic driving or surveillance scenarios, clutter, occlusion, and detection dropout frequently cause identity switches, fragmented trajectories, and missing instances (see missing detections in Fig. 3)—severely degrading the quality of temporally integrated labels. A concurrent work (Skvrna & Neumann, 2025) uses naive mask-based matching without mechanisms for long-range identity preservation, resulting in inconsistent associations. This underscores the need for a more structured and resilient approach to temporal association.

**Why shape incompleteness matters in attribute estimation?** Accurate estimation of 3D attributes—such as object size, orientation, and position—is critical for downstream tasks. Yet existing methods often rely on assumptions that break under open-set detections or noisy observations. For example, object size is typically inferred from category priors, assuming uniform dimensions across instances (Huang et al., 2024b; Zhang et al., 2024a), and orientation from PCA over 'partial point clouds' (Peng et al., 2022b; Huang et al., 2024b), assuming alignment between the principal axis and the object heading. These heuristics are vulnerable to occlusion, truncation, and sparsity, frequently yielding biased annotations. Rather than relying on large-scale training (Brazil et al., 2023) or augmentation (Parihar et al., 2025; Chang et al., 2024), we advocate video-based point aggregation as a more principled and assumption-light alternative.

## 4 METHOD

We propose PLOT, a training-free framework for generating 3D annotations from monocular videos without auxiliary sensors or model retraining. As illustrated in Fig. 3, PLOT leverages off-the-shelf detectors and depth estimators to extract 2D masks and metric depth maps, which are combined via dense point tracking to form temporally grounded correspondences (Sec. 4.1). These are used to estimate relative poses through point-based registration (Sec. 4.2), enabling both shape fusion and motion analysis. To maintain identity consistency and recover missed instances, we introduce a global object association module based on global object memory (GOM) that refines labels over time (Sec. 4.3). Finally, pseudo-LiDARs are constructed by aggregating registered points across frames and projecting them back to the image domain for attribute estimation (Sec. 4.4).

### 4.1 TEMPORAL GROUNDING THROUGH POINT TRACKING

To enable robust object association and temporally grounded geometric reasoning, we distinguish between two types of 2D object mask instances. *Predicted masks* are obtained at each frame via an open-vocabulary detector (GSAM) (Ren et al., 2024), while *tracked masks* are generated by propagating previous-frame instances using a point-based 2D tracker (Harley et al., 2025). The former reflects per-frame shape variations, whereas the latter preserves point-level correspondences necessary for relative pose estimation and shape fusion.

Specifically, given the $k$-th object mask $M_k^t$ from an open-vocabulary 2D detector (Ren et al., 2024) in each frame image $I_t$ where $t \in \mathcal{T}$ (the full set of video time steps), we generate tracked masks

$M_k^{\acute{t}\leftarrow t}$ at other time steps $\acute{t}$ by tracking $M_k^t$ to $\acute{t}$ using a dense point tracker (Harley et al., 2025):

$$M_k^{\acute{t}\leftarrow t} = T_{\acute{t}\leftarrow t}(M_k^t), \quad \acute{t} \in \mathcal{T} \setminus \{t\}, \quad k \in \mathcal{K}^t. \tag{1}$$

Here, $\mathcal{K}^t$ represents the total number of objects predicted at time $t$, and $T_{\acute{t}\leftarrow t}(\cdot)$ denotes dense tracking from $t$ to $\acute{t}$. Note that $\mathcal{K}^t$ can vary between frames due to missing labels caused by distance- or occlusion-related challenges, or simply detector errors.

Thus, the predicted and tracked masks may diverge, despite originating from the same stem. We therefore perform explicit matching to associate them and maintain consistent identity over time. Using frames from two distinct time steps $s$ and $t$, we apply bipartite matching using the Hungarian algorithm (Kuhn, 1955) between the set of tracked masks and the predicted masks, as follows:

$$\hat{\sigma} = \arg\max_{\sigma \in \mathcal{S}_n} \sum_i \text{IoU}(M_i^{s\leftarrow t}, M_{\sigma(i)}^s), \tag{2}$$

where $\hat{\sigma}$ represents the optimal assignment of predicted masks to tracked masks, determined by maximizing the sum of the Intersection of Union (IoU) scores, while $\mathcal{S}_n$ denotes the set of all possible permutations of assignments. Tracked masks are used to estimate rigid motion, while matched predicted masks serve as the basis for shape fusion.

In parallel, we track background points to infer camera motion and disentangle objects from static scene structure. Specifically, we sample background masks $M_{bg}^t$ from regions outside object masks and track them over time as follows:

$$M_{bg}^{\acute{t}\leftarrow t} = T_{\acute{t}\leftarrow t}(M_{bg}^t), \quad \acute{t} \in \mathcal{T} \setminus \{t\}, \quad M_{bg}^t \subset I_t \setminus M^t, \quad M^t = \bigcup_{k \in \mathcal{K}^t} M_k^t. \tag{3}$$

These object and background correspondences are used to determine the motion and dynamic states of both the camera and observed objects, as demonstrated below.

### 4.2 Pose and Motion Estimation via Point Trajectories

In addition to object shape fusion, which relies on relative pose estimation across frames, it is equally important to recover the motion states of both the camera and objects. Accurate motion estimation enables downstream tasks such as object orientation reasoning, trajectory prediction, and relative positioning. To achieve this, we track the trajectories of both background points and object points across consecutive frames. Background point trajectories provide constraints for robust ego-motion estimation, while object point trajectories capture individual object dynamics.

**Camera motion estimation via background trajectories.** To estimate the camera motion, we leverage the trajectories of the background points, which are lifted to 3D using monocular depth predictions (Piccinelli et al., 2024). This yields a set of 3D motion trajectories $\{\mathbf{p}_i^t | i \in M_{bg}^t, t \in \mathcal{T}\}$. The camera motion between a reference frame $r$ (typically the first frame) and each frame $t$ is parameterized by rotation $\mathbf{R}_c^t \in SO(3)$ and translation $\mathbf{t}_c^t \in \mathbb{R}^3$. We recover this motion via Procrustes alignment (Luo & Hancock, 1999):

$$\arg\min_{s_c^t \mathbf{R}_c^t, \mathbf{t}_c^t} \sum_{i \in M_{bg}^t} \mathcal{M} \, \mathcal{V} \, ||\mathbf{p}_i^r - (s_c^t \mathbf{R}_c^t \mathbf{p}_i^{r\leftarrow t} + \mathbf{t}_c^t)||^2, \tag{4}$$

where $\mathcal{M}$ is a binary mask that suppresses background points with unreliable depth estimates (e.g., beyond $50\,\text{m}$), and $\mathcal{V}$ indicates mutual visibility of point $i$ in both frames $t$ and $r$. The scale term $s_c^t$ is optional: it is activated only when monocular depth predictions exhibit noticeable temporal scale drift or flicker; otherwise, we fix $s_c^t = 1$ and recover a purely rigid transformation. This flexibility allows PLOT to remain robust across different depth estimators while avoiding unnecessary degrees of freedom when depth is already stable. The estimated camera motion provides a temporally consistent reference frame, which is later used to refine object poses by compensating for ego-motion.

**Object pose and motion estimation via object trajectories.** Similarly to camera motion, we estimate object motion by computing frame-to-frame registration using reliable point correspondences within each object mask $M_k^t$ and its tracked counterpart $M_k^{r\leftarrow t}$:

$$\arg\min_{s^t \mathbf{R}^t, \mathbf{t}^t} \sum_{i \in M_k^t} \mathcal{V} \, ||\mathbf{p}_i^r - (s^t \mathbf{R}^t \mathbf{p}_i^{r\leftarrow t} + \mathbf{t}^t)||^2, \tag{5}$$

where $\mathbf{R}^t$, $\mathbf{t}^t$ and $s^t$ denote the object's rotation, translation, and scale (optional). The resulting rigid or similarity transformation is later used for temporal aggregation and shape fusion Sec. 4.4.

Given the estimated camera motion by Eq. 4, each object's local trajectory is transformed into the reference frame's coordinate system to obtain a world-aligned trajectory:

$$\hat{\mathbf{p}}_i^t = s_c^t \mathbf{R}_c^t \mathbf{p}_i^t + \mathbf{t}_c^t. \tag{6}$$

The motion status of an object is then determined by its displacement between adjacent frames, i.e., $||\hat{\mathbf{p}}_i^s - \hat{\mathbf{p}}_i^t||$. If the displacement is negligible, the object is treated as static; otherwise, it is considered dynamic. For dynamic objects, orientation is inferred from the direction of motion in world space. Since driving scenes (e.g., KITTI (Geiger et al., 2012)) predominantly exhibit motion along the ground plane, we compute the azimuth angle from trajectory changes:

$$\theta_{k,\text{dyna}}^t = \arctan 2 \left( \frac{\hat{\mathbf{p}}_i^s(x) - \hat{\mathbf{p}}_i^t(z)}{\hat{\mathbf{p}}_i^s(x) - \hat{\mathbf{p}}_i^t(x)} \right). \tag{7}$$

For static objects, we estimate orientation using principal component analysis (PCA) on the fused pseudo-LiDAR from Sec. 4.4, assuming the major axis (highest variance) aligns with the object's longitudinal axis. Unlike prior work (Huang et al., 2024b), our method benefits from denser pseudo-LiDAR from multiple observations, enabling PCA to yield more precise orientation estimates. Per-frame orientation estimates are derived by correcting its orientation based on ego-motion as follows:

$$\theta_{k,\text{static}}^t = \Phi^{-1}\left( (\mathbf{R}_c^t)^{-1} \Phi(\theta_k^r) \right), \tag{8}$$

where $\Phi$ denotes the mapping from a yaw angle to a rotation matrix, and $\Phi^{-1}$ its inverse.

### 4.3 GLOBAL OBJECT MEMORY

Although modern 2D point trackers (Harley et al., 2025; Karaev et al., 2025) and detectors (Ren et al., 2024) perform well per frame, occlusions and clutter still cause missed detections and visibility gaps, leading to fragmented trajectories. These issues manifest as identity switches and noisy pseudo-LiDARs from incorrect point aggregation. To address this, we analyze common failure modes and introduce a Global Object Memory (GOM) that maintains persistent object hypotheses and enforces consistent associations throughout the video.

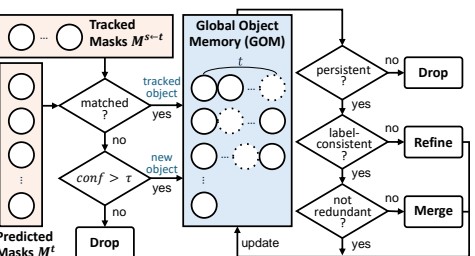

Figure 4: **Global object association pipeline.** Our memory-based object association consolidates noisy frame-level predictions into globally consistent tracks.

**Failure modes in video object tracking.** To examine the limitations of naive tracking-based video understanding, we analyze two dominant failure modes in real-world videos. First, detection dropouts and label noise arise under occlusion, distance, or challenging conditions, where detectors may drop masks or assign inconsistent labels to the same object over time. Second, identity switches in cluttered scenes occur when multiple objects enter, exit, or overlap within a frame, confusing local matching (Eq. 2). In addition, temporary occlusions often reduce the visibility of tracked points below the tracker's confidence threshold, causing the pipeline to prematurely terminate the trajectory-even though the object remains present-resulting in fragmented tracks. Under these conditions, naive mask-level associations tend to assign new identities to previously observed objects or, conversely, merge distinct objects into a single track. Such failure cases highlight the limits of local heuristics and motivate global, temporally structured association; qualitative examples and further analysis are provided in the supplementary material.

**Label refinement via global object memory.** To overcome the limitations of local association, we introduce a global object memory (GOM) that consolidates frame-level predictions into coherent object tracks across the video. As illustrated in Fig. 4, GOM maintains persistent entries, each representing a unique object observed over time. Predicted masks $M^t$ are matched to memory entries based on spatial overlap with the tracked masks $M^{s \leftarrow t}$; if no match is found, a new entry is provisionally created only if the prediction confidence is greater than the detection threshold $\tau$.

Table 1: **Detection results for the 'Car' class on the KITTI (Geiger et al., 2012) validation set.** 'Attr-Net': an additional attribute estimation network. 'Depth': ground-truth depth map used in training. 'Depth†': KITTI-trained depth completion model. 'VPS': Video, Pose and Shape prior. Both supervised and open-set M3OD models use 3D ground-truth data of KITTI in their training.

| | Method | Labels | Extra | $\text{AP}_{\text{3D}}$@IoU = 0.3↑ | | | $\text{AP}_{\text{BEV}}$@IoU = 0.3↑ | | | $\text{AP}_{\text{3D}}$@IoU = 0.5↑ | | | $\text{AP}_{\text{BEV}}$@IoU = 0.5↑ | | |
| | | | | Easy | Mod. | Hard | Easy | Mod. | Hard | Easy | Mod. | Hard | Easy | Mod. | Hard |
|---|---|---|---|---|---|---|---|---|---|---|---|---|---|---|---|
| Sup. | DEVIANT (Kumar et al., 2022) | GT-3D | ✗ | - | - | - | - | - | - | 61.00 | 46.00 | 40.18 | 65.28 | 49.63 | 43.50 |
| | MonoDETR (Zhang et al., 2023) | GT-3D | ✗ | 79.72 | 65.87 | 58.83 | 80.30 | 67.16 | 59.54 | 68.05 | 48.42 | 43.48 | 72.34 | 51.97 | 46.94 |
| Opn. | OVMono3D (Yao et al., 2024) | GT-3D | ✗ | 74.01 | 51.25 | 42.40 | 76.93 | 53.85 | 44.89 | 51.23 | 33.09 | 27.24 | 55.56 | 36.47 | 30.25 |
| | 3D-MOOD (Yang et al., 2025) | GT-3D | Depth | 81.97 | 64.16 | 54.36 | 83.04 | 66.69 | 56.79 | 60.74 | 43.81 | 36.95 | 64.93 | 47.22 | 40.00 |
| Weak | MonoGRNet (Qin et al., 2021) | GT-2D | Attr-Net | 56.16 | 42.61 | 35.36 | 58.61 | 48.75 | 41.49 | 25.66 | 21.57 | 17.40 | 32.23 | 26.88 | 22.47 |
| | WeakM3D (Peng et al., 2022b) | GT-2D | LiDAR | 78.44 | 56.42 | 45.81 | 81.17 | 59.87 | 48.98 | 50.16 | 29.94 | 23.11 | 58.20 | 38.02 | 30.17 |
| | WeakMono3D (Tao et al., 2023) | GT-2D | Stereo | - | - | - | - | - | - | 49.37 | 39.01 | 36.34 | 54.32 | 42.83 | 40.07 |
| | SKD-WM3D (Jiang et al., 2024a) | GT-2D | Depth† | - | - | - | - | - | - | 50.21 | 41.57 | 36.92 | 55.47 | 44.35 | 41.86 |
| Pseudo | OVM3D-Det (Huang et al., 2024b) | GSAM | GPT-4 | 44.48 | 33.29 | 26.69 | 47.42 | 35.96 | 27.99 | 25.63 | 18.85 | 15.67 | 34.52 | 24.46 | 20.40 |
| | MonoSOWA (Skvrna & Neumann, 2025) | MViT2 | VPS | 72.70 | 56.30 | 47.70 | 73.38 | 57.23 | 48.59 | 51.55 | 37.09 | 33.15 | 59.76 | 44.08 | 36.99 |
| | **PLOT (Ours)** | GSAM | Video | **80.48** | **60.83** | **51.49** | **83.06** | **63.59** | **54.15** | **52.02** | 36.78 | 30.40 | **60.25** | 42.80 | 35.77 |

To maintain coherence, GOM updates its object entries by checking 1) whether the object persists across frames, 2) whether the class labels and geometric attributes remain consistent, and 3) whether multiple memory entries correspond to the same object. Based on these checks, GOM either 1) discards unstable or incoherent instances, 2) refines entries by updating their attributes with dominant evidence, or 3) merges redundant entries. This refinement enforces long-term identity consistency and improves the reliability of object-level 3D annotations.

In summary, GOM addresses two complementary sources of discontinuity: 1) its primary role is to restore continuity when the detector produces missed or duplicated masks, and 2) it reconnects trajectory segments that become separated when occlusion reduces point visibility and interrupts the tracker's propagation.

## 4.4 TRAJECTORY-GUIDED OBJECT SHAPE FUSION

Using the globally associated object tracks constructed in Sec. 4.3, we build complete pseudo-LiDARs by aggregating object observations over time. Rather than requiring absolute camera poses, we perform shape fusion using relative poses derived from point trajectories in Sec. 4.2.

For each object, we first identify the reference frame image $I_r$ that minimizes the total registration error between matched frames, using Eq. 5. We then align all matched object masks to this target frame using their estimated relative rotation and translation:

$$\hat{\mathbf{P}}_k = \bigcup_{t \in \mathcal{T}} s^{t \to r} \mathbf{R}^{t \to r} \mathbf{P}_k^t + \mathbf{t}^{t \to r}, \tag{9}$$

where $\mathbf{P}_k^t$ denotes the 3D points extracted from the $k$-th object mask at time $t$, and $t \to r$ indicates the transformation from $t$ to the reference frame $r$. The resulting point cloud $\hat{\mathbf{P}}_k$ serves as a completed pseudo-LiDAR representation, fusing multiple partial views into a consistent shape.

To ensure reliable 3D annotations across all frames, we further project the completed pseudo-LiDAR $\hat{\mathbf{P}}_k$ back into each frame's coordinate system using the inverse of the same relative transformation.

## 5 EXPERIMENTS

In this section, we evaluate PLOT on standard monocular 3D object detection (M3OD) video benchmarks—KITTI (Geiger et al., 2012), KITTI-360 (Liao et al., 2022), and Waymo (Sun et al., 2020)—and compare it against recent open-set detectors, weakly-supervised and pseudo-labeling methods that rely on driving-specific priors or pose estimates. While our labeler is designed for open-world settings, existing benchmarks with video inputs are constrained to driving scenes, limiting the scope of evaluation. Thus, we provide qualitative comparisons on in-the-wild videos (Ding et al., 2023; Milan et al., 2016; Contributors, 2025) to assess the generalization of PLOT beyond vehicle-centric environments. Finally, we present ablations to validate each component, with further details and results in the appendix.

## 5.1 EXPERIMENTS ON DRIVING BENCHMARKS

In this subsection, we compare our method with various M3OD works (listed on the left of each table) on multiple benchmark datasets. The methods are grouped as **Sup.** (fully-supervised), **Weak.** (weakly-supervised), **Open.** (open-set), and **Pseudo.** (pseudo-labeling). For pseudo-labeling methods, including ours, results are obtained by training MonoDETR (Zhang et al., 2023) on

Table 2: **Detection results for the 'Car' class on the KITTI-360 (Liao et al., 2022) test set.**

| | Method | Labels | Extra | $AP_{3D}$@0.3↑ Easy | Hard | $AP_{BEV}$@0.3↑ Easy | Hard |
|---|---|---|---|---|---|---|---|
| Sup. | MonoDETR (CVPR'23) | GT-3D | ✗ | 64.32 | 57.83 | 66.42 | 60.31 |
| Open. | OVMono3D (arXiv'24) | Zero-shot | ✗ | 41.82 | 31.94 | 49.45 | 39.08 |
| | 3D-MOOD (ICCV'25) | Zero-shot | ✗ | 21.15 | 13.12 | 32.68 | 21.41 |
| Weak. | Autolabels (CVPR'20) | GT-2D | LiDAR, Shape | 12.92 | 9.94 | 48.16 | 37.34 |
| | WeakM3D (ICLR'22) | GT-2D | LiDAR | 21.25 | 15.34 | 29.89 | 24.01 |
| Pseudo. | VSRD (CVPR'24) | GT-2D | Pose | 50.86 | 43.45 | 58.40 | 50.61 |
| | MonoSOWA (ICCV'25) | MViT2 | VPS | 42.72 | 46.59 | 50.84 | 49.22 |
| | **PLOT (Ours)** | GSAM | Video | **54.78** | **48.75** | **60.75** | **54.58** |

the generated labels with the same default configuration across all methods to ensure fairness; additional details are provided in the appendix. As most M3OD methods focus on the 'Car' class, our main paper reports quantitative results primarily for this category. Additional quantitative results on other classes (e.g., 'Pedestrian') and qualitative results are provided in the appendix.

Table 3: **Detection results for the 'Vehicle' class on the Waymo (Sun et al., 2020) validation set.**

| | Method | Labels | Extra | $AP_{3D}$@0.5↑ All | 0-30m | 30-50m | 50-∞m | $AP_{BEV}$@0.5↑ All | 0-30m | 30-50m | 50-∞m |
|---|---|---|---|---|---|---|---|---|---|---|---|
| Sup. | DEVIANT (Kumar et al., 2022) | GT-3D | ✗ | 10.29 | 26.75 | 4.95 | 0.16 | - | - | - | - |
| | CaDDN (Reading et al., 2021) | GT-3D | LiDAR | 16.51 | 44.87 | 8.99 | 0.58 | - | - | - | - |
| | MonoDETR (Zhang et al., 2023) | GT-3D | ✗ | 21.41 | 37.89 | 18.61 | 3.69 | 23.63 | 39.10 | 20.95 | 4.70 |
| Open. | OVMono3D (Yao et al., 2024) | Zero-shot | ✗ | 5.46 | 14.68 | 5.61 | 0.65 | 6.11 | 15.24 | 6.54 | 0.99 |
| | 3D-MOOD (Yang et al., 2025) | Zero-shot | ✗ | 1.13 | 2.44 | 1.68 | 0.05 | 4.60 | 9.80 | 6.71 | 0.30 |
| Weak. | WeakM3D (Peng et al., 2022b) | GT-2D | LiDAR | 4.50 | 12.16 | 3.67 | 0.40 | 8.18 | 20.31 | 7.50 | 0.99 |
| Pseudo. | MonoSOWA (Skvrna & Neumann, 2025) | MViT2 | VPS | 13.46 | 24.65 | 5.87 | 4.55 | 18.98 | 33.51 | 12.02 | 4.55 |
| | **PLOT (Ours)** | GSAM | Video | **14.04** | **26.97** | **17.71** | **4.87** | **23.32** | **41.74** | **31.39** | **8.22** |

**KITTI & KITTI-360.** Tab. 1 and Tab. 2 provide quantitative results on the KITTI and KITTI-360 datasets for the 'Car' class. For clarity, the label column lists both 2D ground-truth annotations (GT-2D) and predictions from 2D detectors, MVit2 (Li et al., 2022) and GSAM (Ren et al., 2024), used in place of ground truth. On KITTI, PLOT outperforms the single-image pseudo-labeling baseline OVM3D-Det by an average of +19 AP, and also surpasses weakly-supervised methods that rely on auxiliary sensors such as stereo (WeakMono3D) and LiDAR (WeakM3D). PLOT also achieves results comparable to fully supervised baselines and open-set detectors. Compared to the concurrent work MonoSOWA, which relies on IMU sensors and simplified shape assumptions, PLOT achieves superior AP@0.3 (up to +9.68) and, as illustrated in Fig. 5, yields more accurate 3D boxes by leveraging trajectory-guided shape fusion and global object association. On KITTI-360, PLOT also improves over recent pseudo-labeling methods such as VSRD (Liu et al., 2024) and MonoSOWA, achieving the best AP@0.3 in both Easy and Hard regimes.

**Waymo-Open.** We report results on the Waymo-Open dataset in Tab. 3. Unlike KITTI benchmarks, Waymo features long, diverse sequences under adverse conditions such as night and rain,

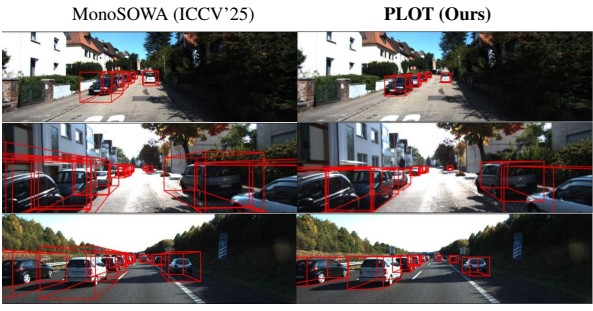

Figure 5: **Qualitative comparisons on KITTI.** Only the predicted boxes are displayed for clarity.

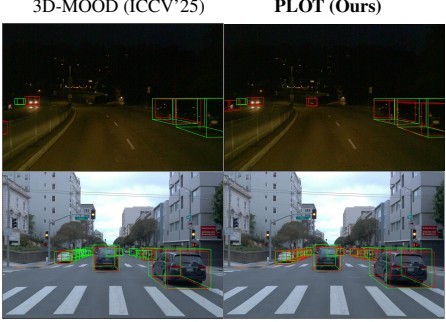

Figure 6: **Qualitative comparison on Waymo-Open.**

making it a strong proxy for real-world complexity. PLOT achieves the best performance in distant regions (>30m), surpassing fully supervised baselines, and shows superior $AP_{BEV}$ across nearly all ranges. As shown in Fig. 6, PLOT remains effective under challenging conditions such as night scenes, even recovering valid instances missing from the ground truth (red: predictions, green: ground truth), underscoring the scalability and reliability of our framework in unconstrained environments.

## 5.2 EXPERIMENTS ON IN-THE-WILD VIDEOS

To demonstrate the open-set capability of our method, we present qualitative comparisons on in-the-wild videos using identical text prompts (e.g., "boat") as inputs for PLOT, state-of-the-art open-set method, 3D-MOOD (Yang et al., 2025), and single-image pseudo-labeling method OVM3D-Det (Huang et al., 2024b); MonoSOWA (Skvrna & Neumann, 2025), which depends on auxiliary sensors and object shape priors, is not applicable here. As shown in Fig. 2 and Fig. 7, PLOT generalizes well under diverse and challenging conditions (e.g., variations in scene layouts, camera setups) while maintaining accurate object attribute estimates. In contrast, 3D-MOOD does not detect any objects in the 'boat' scene (left) in Fig. 7, and OVM3D-Det of-

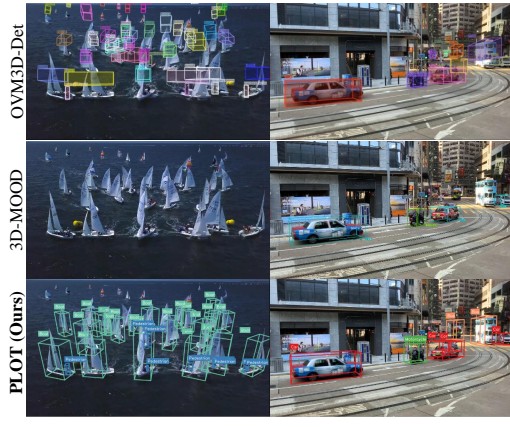

Figure 7: **Qualitative comparisons in the wild.**

ten produces noisy estimates with most falls back to priors of LLM-driven fixed objects, due to single image observations. Such in-the-wild evaluations are particularly important, as they go beyond structured driving benchmarks and validate the scalability of our method to unconstrained real-world settings. Additional qualitative results are provided in the appendix, with video demonstrations in the supplementary material.

Table 4: **Ablation on trajectory-guided object shape fusion and number of adj. frames.**

| #Fr. | $AP_{3D}@IoU = 0.3\uparrow$ | | | $AP_{BEV}@IoU = 0.3\uparrow$ | | | Time |
|---|---|---|---|---|---|---|---|
| | Easy | Mod. | Hard | Easy | Mod. | Hard | (10/20 obj.) |
| 1 | 51.75 | 37.48 | 30.63 | 57.41 | 41.96 | 34.00 | **0.46 / 0.47** |
| +2 | 62.41 | 48.06 | 40.31 | 67.57 | 51.39 | 43.35 | 0.66 / 0.70 |
| +8 | 77.23 | 56.16 | 48.43 | 80.27 | 59.13 | 51.29 | 1.56 / 1.73 |
| +20 | **80.48** | **60.83** | **51.49** | **83.06** | **63.59** | **54.15** | 2.45 / 2.83 |

Table 5: **Ablation on global object memory and PLOT as standalone detector.**

| | Train | GOM | $AP_{3D}@IoU = 0.3\uparrow$ | | | $AP_{BEV}@IoU = 0.3\uparrow$ | | |
|---|---|---|---|---|---|---|---|---|
| | | | Easy | Mod. | Hard | Easy | Mod. | Hard |
| KIT. | ✗ | ✓ | 59.82 | 51.00 | 45.40 | 62.41 | 54.18 | 48.44 |
| | ✓ | ✓ | **80.48** | **60.83** | **51.49** | **83.06** | **63.59** | **54.15** |
| K360 | ✓ | ✗ | 48.14 | - | 40.28 | 54.86 | - | 46.61 |
| | ✓ | ✓ | **54.78** | - | **48.75** | **60.75** | - | **54.58** |

## 5.3 ABLATION STUDIES

**Efficacy of trajectory-guided shape fusion.** Tab. 4 compares trajectory-guided shape fusion with varying frame counts, reporting detection performance and labeling time per frame on KITTI. Performance improves with more frames, achieving up to $1.7\times$ higher average AP compared to single-frame labels. The rightmost column of the table shows the computation time of the whole pipeline, which grows proportionally with the number of objects, yet remains practical (running at $2.83$s per frame on a single CPU with a RTX 3090 GPU) with 20 objects per scene, while the KITTI dataset on average contains about 10–12 objects per frame. Considering both accuracy and computation time, we adopt a 20-frame window for the experiments.

**PLOT as standalone 3D detector.** In the top 2 rows of Tab. 5, we examine the feasibility of using PLOT as a standalone 3D detector without training an additional M3OD model on KITTI. Remarkably, the raw pseudo-labels already surpass the previous pseudo-labeling method (OVM3D-Det) and the weakly-supervised method (MonoGRNet) (see Tab. 1), suggesting the utility of PLOT in scenarios where training is infeasible, such as in-the-wild videos or cases with only a single sequence available. However, the integration of uncertainty-aware depth estimation and 2D–3D geometric consistency in MonoDETR mitigates depth errors and label inconsistencies, yielding more stable and accurate 3D detections. Further analysis can be found in the appendix.

Table 6: **Ablation on the impact of tracker and depth noises.**

| Alignment | Range Noise | Patch Noise | Temporal Jitter | Tiny Tracker | Depth RMSE ↓ | Pose ATE ↓ | AP$_{3D}$@0.3 ↑ Easy | Hard | AP$_{BEV}$@0.3 ↑ Easy | Hard |
|---|---|---|---|---|---|---|---|---|---|---|
| SE(3) | - | - | - | ✗ | **3.72** | **0.204** | **20.89** | **14.38** | **27.85** | **20.20** |
| SE(3) | - | - | - | ✓ | **3.72** | **0.204** | 20.57 (-0.32) | 14.17 (-0.21) | 27.71 (-0.14) | 19.54 (-0.66) |
| SE(3) | 0.04 | 0.3 | ✗ | - | 3.90 (+0.18) | 0.205 | 20.30 (-0.59) | 14.00 (-0.38) | 28.80 (+0.95) | 19.73 (-0.47) |
| | 0.08 | 0.3 | ✗ | - | 4.34 (+0.62) | **0.204** | 19.48 (-1.41) | 13.45 (-0.93) | 26.63 (-1.22) | 19.35 (-0.85) |
| | 0.08 | 0.6 | ✗ | - | 4.38 (+0.66) | 0.206 | 19.65 (-1.24) | 13.39 (-0.99) | 26.94 (-0.91) | 19.39 (-0.81) |
| SE(3) | ✗ | ✗ | 0.02 | - | 3.78 (+0.06) | 0.205 | 17.50 (-3.39) | 11.96 (-2.42) | 25.84 (-2.01) | 17.43 (-2.77) |
| | | | 0.04 | - | 3.82 (+0.10) | **0.204** | 14.14 (-6.75) | 9.49 (-4.89) | 20.29 (-7.56) | 14.54 (-5.66) |
| Sim(3) | ✗ | ✗ | 0.02 | - | 3.78 (+0.06) | **0.204** | 17.69 (-3.20) | 13.39 (-0.99) | 25.96 (-1.89) | 19.01 (-1.19) |
| | | | 0.04 | - | 3.81 (+0.09) | **0.204** | 14.99 (-5.90) | 11.23 (-3.15) | 22.55 (-5.30) | 16.36 (-3.84) |

**Necessity of global object memory.** We evaluated the impact of global object memory on KITTI-360 (K360) in the bottom 2 rows of Tab. 5, as KITTI's 3D detection benchmark is not suitable for assessing GOM, whereas KITTI-360 provides continuous sequences. The capability of GOM to reduce 2D detector-induced noise and maintain temporal consistency results in more stable and accurate pseudo-labels. Such video-consistent label generation is directly relevant for practical annotation pipelines and downstream 3D perception tasks. Our video-consistent label results can also be viewed in the Waymo video demonstration provided in the supplementary material.

**Robustness to tracking quality.** As part of the stage-wise analysis summarized in Tab. 6, we evaluate how variations in tracking accuracy influence downstream labeling quality. To this end, we replace the standard AllTracker (Harley et al., 2025) with a tiny, less accurate but faster variant and measure its effect on camera motion, fused geometry, and 3D attributes. As shown in the first row of Tab. 6, the overall performance drops only mildly despite the reduced tracking accuracy, indicating that global object memory effectively compensates for fragmented trajectories and helps preserve temporal consistency. However, adopting a more robust tracking model could further enhance PLOT's labeling performance.

**Robustness to depth noise and temporal scale inconsistencies.** Since PLOT relies on monocular depth, one may question its stability under depth noises, such as spatial noises and temporal scale fluctuations. To assess this, we inject controlled perturbations into the estimated depth on a sampled Waymo-Open (Sun et al., 2020) scene and evaluate depth quality, camera motion, and labeling accuracy. As shown in Tab. 6, camera motion estimation remains stable due to reliable background correspondences. Depth error grows mainly under range-dependent noise, while spatial noise causes only mild degradation in shape fusion and 3D attributes. In contrast, non-uniform temporal scale noise leads to larger drops in box AP, reflecting a depth sensitivity shared with other monocular-depth–based pipelines and indicating that advances in depth estimation will naturally improve PLOT. The last two rows show results with Sim(3) alignment (introducing the optional scale term in Eq.4 and 5), which improves the box AP scores with stronger temporal noise. Furthermore, as shown in the supplemental video and the frame-wise BEV visualization in the appendix, modern depth estimators exhibit minimal flickering or inter-frame scale drift in large-scale scenes.

## 6 CONCLUSION

In this paper, we introduced PLOT, a framework for generating reliable 3D annotations from monocular videos through tracking-driven object association and label refinement. Beyond monocular 3D detection, this video-based formulation holds potential for broader 3D tasks that suffer from missing camera information or incomplete shapes, such as CAD model retrieval and object-level reconstruction. We hope that this paradigm provides a scalable alternative to data-intensive monocular 3D detection pipelines and opens new directions for video-based 3D understanding without explicit supervision.

**Limitations.** Like most open-set 3D perception frameworks, PLOT relies on pre-trained depth estimators. While this is not unique to our approach, depth errors under noise or at long ranges remain the main bottleneck. In addition, orientation for static objects remains inherently ambiguous under single-view observations, where we adopt PCA as a practical fallback. While future advances may mitigate these issues, we provide further analysis in the appendix, including additional failure modes.

REPRODUCIBILITY STATEMENT

The appendix specifies the hardware and software configurations used in all experiments, together with implementation details and hyperparameters. Our method builds on explicit shape fusion for pseudo-labeling, which avoids heuristic parameter tuning. We report quantitative and qualitative results across multiple video-based M3OD benchmarks and zero-shot in-the-wild videos. For baselines, we use official results when available and otherwise reproduce them with publicly released code under recommended settings. Partial code and video demonstrations are provided in the supplementary material, and the full code will be released upon acceptance. The released code will include labeling and evaluation scripts for all benchmark datasets as well as for zero-shot video inputs. Our method, despite its strong generalizability, does not require massive computational resources, further facilitating reproducibility.

ETHICS STATEMENT

This work makes use of publicly available datasets and videos, including material from Pexels under its permissive license. While the data are licensed for public research use, some videos may contain identifiable individuals. We acknowledge that such content raises potential privacy concerns and encourage responsible use in line with community standards. Our method does not attempt to infer personal identity or sensitive attributes, and the broader aim is to advance scalable video-based 3D perception for autonomous systems and robotics. On the positive side, by lowering the cost of large-scale 3D annotation, our approach can help democratize access to high-quality training data, thereby broadening participation in 3D perception research. We believe this work poses minimal risk but emphasize the importance of careful dataset curation and respect for privacy in downstream applications.

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

# APPENDIX

This appendix complements the main content of the article by improving reproducibility and providing additional evidence of efficacy. We first describe the details of the method required to reproduce our approach (A), followed by the experimental details needed for comparisons with baselines (B). We then provide qualitative examples that illustrate the motivation for our method (C), followed by additional experiments that clarify the efficacy of our method and the role of each component (D). We also present a dedicated analysis of failure cases (E), which summarizes the main limitations and discusses remaining challenges. Finally, we show qualitative results that highlight aspects of our approach not fully reflected by quantitative metrics (F), and describe the usage of large language models (G).

In addition to this appendix, the supplementary file contains two components: (i) partial source code for reproducibility, and (ii) video-consistent demos. The demos include several in-the-wild videos, which illustrate the frame-to-frame consistency of pseudo-labeling without training, and video demonstrations on some of the Waymo (Sun et al., 2020) scenes, which visualize predictions from a model trained with our pseudo-labels. Both types of videos further demonstrate the stability of our approach.

## A   METHOD DETAILS

**Usage of foundation models.**   For pseudo-label generation, we leverage three foundation models: Grounded-SAM (Ren et al., 2024) for 2D detection and segmentation, AllTracker (Harley et al., 2025) for dense point tracking, and UniDepth (Piccinelli et al., 2024) or MoGe2 (Wang et al., 2025c) for metric depth estimation. For 2D label generation with Grounded-SAM (Ren et al., 2024), we use the prompts with the categories: 'Car'/'Vehicle' and 'Pedestrian' for driving benchmarks and adapt relevant class labels for in-the-wild videos, with a box threshold of $0.3$ and a text threshold of $0.25$. To ensure fairness, we adopt the same text prompts as used in other prompt-based baselines (Yao et al., 2024; Yang et al., 2025; Huang et al., 2024b). For depth estimation, we use ground-truth camera intrinsics on driving benchmarks and estimated intrinsics from each depth model for in-the-wild videos. We find UniDepth (Piccinelli et al., 2024) to be highly accurate in driving scenarios, and therefore adopt it for experiments on driving benchmarks, while MoGe2 (Wang et al., 2025c) is used for in-the-wild videos. This modular use of foundation models facilitates the adaptation of PLOT across domains. Our choice of 2D detector and depth estimator follows OVM3D-Det (Huang et al., 2024b), where as MonoSOWA (Skvrna & Neumann, 2025) uses COCO-trained MViT-v2 (Li et al., 2022) detector and Metric3Dv2 (Hu et al., 2024) depth estimator.

**Hyperparameters for pseudo-labeling.**   Pseudo-labels are generated on every 20-frame tracking window (see Tab. 4, main paper), where dense tracks are computed jointly across all frames. For longer sequences in KITTI-360 (Liao et al., 2022) and Waymo (Sun et al., 2020), videos are split into non-overlapping windows of this length. Object visibility, which is used in our camera-motion estimation (Sec. 4.2; Eq. (4)), is computed from point-track visibility scores with a threshold of $0.6$. Objects that remain unmatched for $4$ consecutive frames are considered to have left the scene, while tracklets shorter than $5$ frames are discarded. However, these thresholds may be adjusted according to the video frame rate. The IoU threshold between a tracklet and its corresponding detection, used in the Hungarian Matching (Kuhn, 1955) in Eq. (2) of Sec. 4.1, is set to $0.4$. During camera motion estimation, we filter points with depth greater than $50$ m, while object motion estimation uses all available points without depth filtering. An object is considered moving if its displacement, i.e., $||\hat{\mathbf{p}}_i^s - \hat{\mathbf{p}}_i^t||$, exceeds the threshold of $2$ m.

**MonoDETR training.**   Using the pseudo-labels generated on the training split, we train MonoDETR (Zhang et al., 2023), a single-image monocular 3D detector, in a fully supervised manner. Its performance is then evaluated on the corresponding validation or test split, depending on the dataset. All training hyperparameters follow the official setting without modification.

**Software specification.**   A list of dependencies and versions of used libraries is provided in the `requirements.txt` of the supplementary codes.

## B EXPERIMENTAL DETAILS

**Hardware specification.** Pseudo-label generation and the run-time analysis reported in Tab. 4 of the main paper are conducted on a PC equipped with an Intel Core i9-14900KF CPU and a single NVIDIA RTX-3090 GPU. The training of MonoDETR (Zhang et al., 2023) is performed separately on a PC with a single NVIDIA A100 GPU.

**Benchmark datasets.** We evaluate PLOT on three monocular 3D object detection (M3OD) benchmarks with video sequences: KITTI (Geiger et al., 2012), KITTI-360 (Liao et al., 2022), and Waymo Open (Sun et al., 2020). For KITTI, we adopt the official training and validation split of the 3D object detection benchmark, consisting of 3,712 training images and 3,769 validation images. Since the KITTI M3OD benchmark is defined for single images, we leverage adjacent frames from the raw video sequences for each labeled image to enable temporal reasoning, simliar to the concurrent work (Skvrna & Neumann, 2025). For KITTI-360, we follow the protocol in VSRD (Liu et al., 2024), which uses 6 training sequences with 44,178 frames and 1 test sequence with 2,459 frames. For the Waymo Open dataset, we follow the official split, consisting of 798 training sequences (158,080 frames) and 202 validation sequences (39,988 frames).

**In-the-wild videos.** Since all public M3OD benchmarks are limited to driving scenes, we further evaluate generalization by applying PLOT to diverse monocular videos outside this domain. We specifically select datasets that capture scenes rarely covered by standard 3D detection benchmarks, including surveillance footage (MOT17 (Milan et al., 2016), DIVOTrack (Hao et al., 2024)), hand-held recordings (MOSE (Ding et al., 2023)), and crowd-sourced videos (GMOT-40 (Bai et al., 2021), Pexels (Contributors, 2025)). Unlike structured driving environments with forward-facing cameras and narrow fields of view, these in-the-wild settings introduce unconstrained camera motions, varied scene structures, and diverse object categories. Such characteristics make them particularly valuable for assessing robustness under open-set evaluation, where models must handle novel environments without domain-specific assumptions. To illustrate this, we compare results on representative scenarios, including a CCTV-like view (Fig. 2, main paper) and a 'boat' scene (Fig.7, main paper). In these experiments, we use raw pseudo-labels without training and compare them with prior pseudo-labeling methods(Huang et al., 2024b) as well as zero-shot predictions from open-set detectors(Yang et al., 2025), using the same object categories for fairness.

**Weakly-supervised M3OD models.** We compare PLOT against recent weakly-supervised methods (Qin et al., 2021; Peng et al., 2022b; Tao et al., 2023; Jiang et al., 2024a). These approaches avoid full 3D supervision but rely on additional priors, such as separately trained networks or auxiliary inputs (e.g., LiDAR, stereo, or depth). Specifically, MonoGRNet (Qin et al., 2021) trains a separate attribute estimation network to circumvent direct 3D bounding-box regression, while SKD-WM3D (Jiang et al., 2024a) leverages a depth completion network trained on the same benchmark dataset. In addition, these methods typically exploit 2D ground-truth annotations to provide extra cues in the absence of full 3D supervision. In contrast, PLOT eliminates the need for such priors or auxiliary inputs, learning directly from monocular video sequences through pseudo-labeling. All baseline results are taken from officially reported results.

**Open-set M3OD models.** Given the open-vocabulary nature of our approach, we also compare with open-set monocular 3D object detectors (Yao et al., 2024; Yang et al., 2025). These models are trained on large-scale cross-domain datasets with dense 3D ground-truth annotations, such as Omni3D (Brazil et al., 2023), which already include subsets of driving benchmarks - KITTI (Geiger et al., 2012) and nuScenes (Caesar et al., 2020). This overlap makes direct benchmark comparisons less meaningful, but it underscores the practical gap between approaches that depend on curated large-scale 3D supervision and PLOT, which learns from monocular video alone without requiring domain-specific priors or additional annotations.

**Pseudo-labelers.** We primarily compare against previous pseudo-labeling approaches (Huang et al., 2024b; Liu et al., 2024; Skvrna & Neumann, 2025). Some methods introduce generic priors, such as LLM-estimated object sizes (Huang et al., 2024b), while others rely on assumptions tightly coupled to driving domains, such as known sensor poses (Liu et al., 2024; Skvrna & Neumann, 2025) or fine-grained object template shapes (e.g., *minivan*, *Passat*, *SUV*, *sedan*) (Skvrna &

Neumann, 2025). In contrast, PLOT generates pseudo-labels directly from raw monocular videos without requiring such additional priors.

**Evaluation metrics.** For the KITTI and KITTI-360 datasets, we report average precision in 3D and bird's eye view ($\mathrm{AP_{3D}}/\mathrm{AP_{BEV}}$), computed at 40 recall positions and using two IoU thresholds: 0.3 and 0.5. The results are reported in three levels of difficulty: easy, moderate (KITTI only), and hard, based on the size of the object's bounding box. For the Waymo Open dataset, we evaluate on Level_2 objects and use the same metrics ($\mathrm{AP_{3D}}/\mathrm{AP_{BEV}}$) with an IoU threshold of 0.5. The results are further broken down by distance ranges of objects: $[0, 30)$, $[30, 50)$, and $[50, \infty)$ meters.

# C FURTHER ANALYSIS ON MOTIVATION

This section provides qualitative examples and additional explanations that support the motivation in Sec. 3 of the main paper.

**Failures in object association.** Both video-frame tracking and 2D detection are error-prone under occlusion, clutter, and detector noise. As a result, naive box-level matching that relies only on spatial proximity (Skvrna & Neumann, 2025) inevitably produces incomplete associations, ultimately requiring reliance on shape priors and creating a barrier to generalization. The qualitative comparisons in Fig. 8 illustrate how naive tracking leads to association failures: the pseudo-LiDAR of a single object becomes fragmented, resulting in additional false positive boxes. Reliable object association, built on consistent object

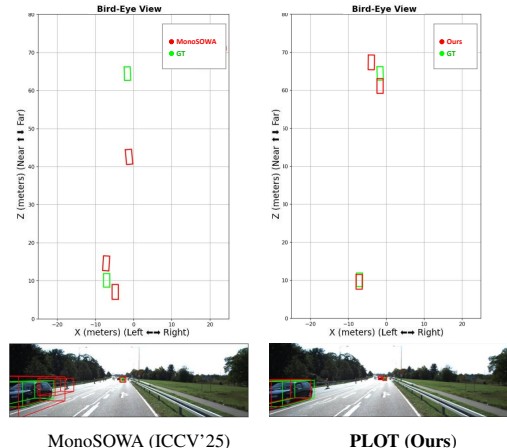

MonoSOWA (ICCV'25)  **PLOT (Ours)**

Figure 8: **Association failures with naive tracking.**

identities and trajectories, is essential for our method, which performs trajectory-guided shape fusion and relative-motion-based orientation estimation. Fig. 9 illustrates this effect by showing BEV locations of detected boxes over time without global object memory (left) and with it (right). When shape fusion is performed based on object associations on the left figure, severe pseudo-LiDAR distortions arise, which in turn lead to errors in subsequent attribute estimation.

**Shape incompleteness in attribute estimation** Attribute estimation in M3OD often fails when derived from incomplete pseudo-LiDARs. Existing approaches typically compute orientation via PCA on the observed points, while dimensions and centers are similarly obtained from the same

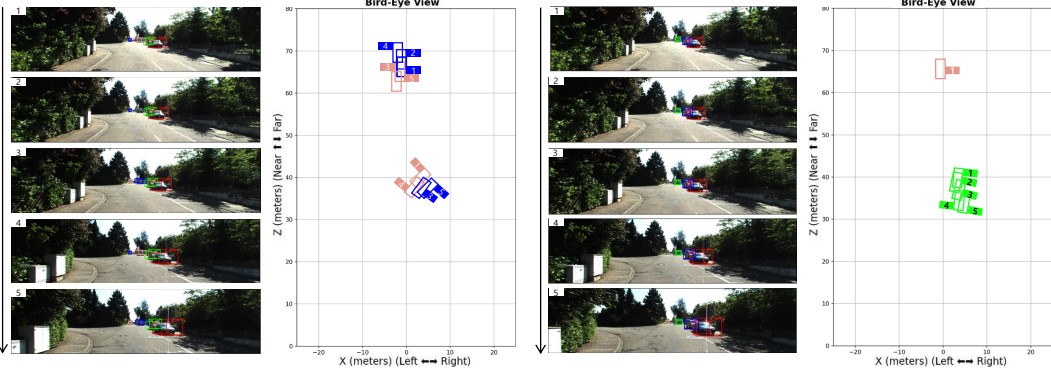

Figure 9: **Effect of accurate object association on identity consistency.** BEV visualizations with and without proper association (GOM) show how ID switches are reduced when associations are corrected. Zoom for the best view.

Figure 10: **Completed pseudo-LiDAR obtained by trajectory-guided shape fusion.** The middle row shows an object's multiple observations across time through ego-motion, which enables reliable orientation and dimension estimation.

Table 7: **Detection results for the 'Pedestrian' class on KITTI (Geiger et al., 2012) val set.**

| | Method | Labels | Extra | AP$_{3D}$@IoU = 0.3↑ Easy | Mod. | Hard | AP$_{BEV}$@IoU = 0.3↑ Easy | Mod. | Hard | AP$_{3D}$@IoU = 0.5↑ Easy | Mod. | Hard | AP$_{BEV}$@IoU = 0.5↑ Easy | Mod. | Hard |
|---|---|---|---|---|---|---|---|---|---|---|---|---|---|---|---|
| Sup. | MonoDIS (Simonelli et al., 2019) | GT-3D | ✗ | - | - | - | - | - | - | 3.20 | 2.28 | 1.71 | 4.04 | 3.19 | 2.45 |
| | MonoXiver (Liu et al., 2023) | GT-3D | ✗ | - | - | - | - | - | - | 7.95 | 5.49 | 4.62 | - | - | - |
| | GUP-Net (Lu et al., 2021) | GT-3D | ✗ | - | - | - | - | - | - | 9.37 | 6.84 | 5.73 | - | - | - |
| | DEVIANT (Kumar et al., 2022) | GT-3D | ✗ | - | - | - | - | - | - | 9.85 | 7.18 | 5.42 | - | - | - |
| W | WeakM3D Peng et al. (2022b) | GT-2D | LiDAR | - | - | - | 3.79 | 3.21 | 3.12 | - | - | - | - | - | - |
| Psd. | OVM3D-Det (Huang et al., 2024b) | GSAM | GPT-4 | 10.63 | 8.96 | 7.32 | 11.25 | 9.36 | 7.85 | 8.19 | 6.88 | 5.56 | 9.11 | 7.71 | 6.27 |
| | **PLOT (Ours)** | GSAM | Video | **16.64** | **14.39** | **12.27** | **17.66** | **15.01** | **12.76** | **13.97** | **11.71** | **9.84** | **14.52** | **12.48** | **10.59** |

shape, e.g., the center is estimated as a mean of pseudo LiDAR $P$, as follows:

$$(x_c, y_c, z_c) = \text{mean}(P). \tag{10}$$

However, when the shape is sparse or truncated, the box center and orientation become shifted or distorted, forcing reliance on class-level priors rather than observed evidence, as observed in the 3D bounding box estimates of OVM3D-Det (Fig. 7, main paper) that uses single-image pseudo-LiDAR. In contrast, as illustrated in Fig. 10, our trajectory-guided shape fusion yields more complete pseudo-LiDARs, making orientation, dimension, and center estimation significantly more robust.

# D  ADDITIONAL EXPERIMENTS

All additional experiments are conducted on the KITTI (Geiger et al., 2012) dataset.

**KITTI-Pedestrian.** Beyond the commonly evaluated 'Car' class in driving benchmarks, we further assess performance on the more challenging 'Pedestrian' class in the KITTI validation split. Detecting pedestrians in monocular 3D is particularly difficult due to their small size, frequent occlusions, and the severe depth ambiguity that arises when estimating 3D attributes from limited visual evidence. As shown in Tab. 7, PLOT surpasses all prior methods, including fully supervised approaches, by a significant margin. This result highlights the effectiveness of trajectory-guided shape fusion in capturing fine-grained object geometry, enabling reliable orientation and localization even for small, highly dynamic objects, such as pedestrians.

Table 8: **Ablation on orientation estimation.**

| Cam motion | AP$_{3D}$@IoU = 0.3↑ Easy | Mod. | Hard | AP$_{BEV}$@IoU = 0.3↑ Easy | Mod. | Hard |
|---|---|---|---|---|---|---|
| ✗ | 77.65 | 57.59 | 48.27 | 80.51 | 60.25 | 50.81 |
| ✓ | **80.48** | **60.83** | **51.49** | **83.06** | **63.59** | **54.15** |

Table 9: **Ablation on estimated intrinsics.**

| Intrinsics | AP$_{3D}$@IoU = 0.3↑ Easy | Mod. | Hard | AP$_{BEV}$@IoU = 0.3↑ Easy | Mod. | Hard |
|---|---|---|---|---|---|---|
| UniDepth | 70.44 | 51.90 | 44.12 | 76.17 | 58.66 | 50.49 |
| GT | **80.48** | **60.83** | **51.49** | **83.06** | **63.59** | **54.15** |

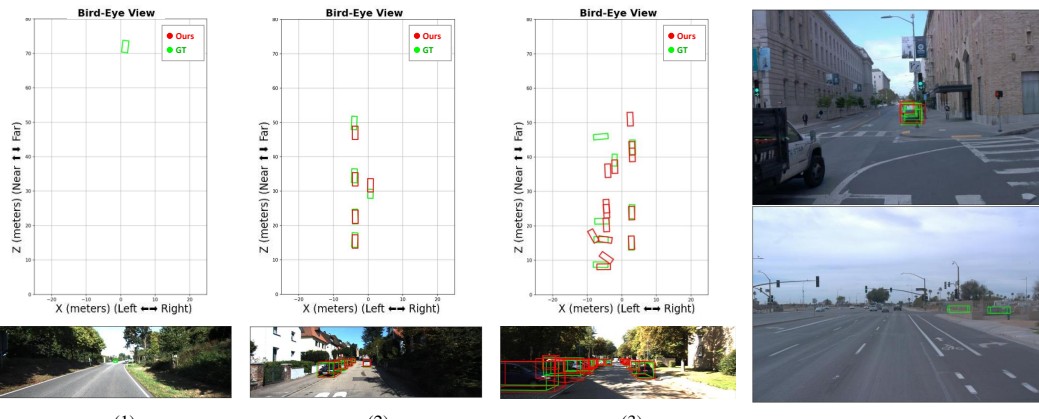

(1)          (2)          (3)

Figure 11: **Failure modes analysis on KITTI (Geiger et al., 2012).**

Figure 12: **Failure cases in Waymo-Open.**

**Motion-guided orientation estimation.** In Tab. 8, we evaluate the role of camera motion (i.e., relative camera poses) in transforming local object trajectories into world-space alike trajectories. Without explicit motion reasoning to distinguish between static and dynamic objects, orientation estimation becomes less reliable, leading to a noticeable drop in performance.

**Estimated intrinsics.** In Tab. 9, we evaluate PLOT using UniDepth-estimated camera intrinsics in place of ground-truth values. Although intrinsic calibration is typically available in driving benchmarks, the results indicate that PLOT maintains performance comparable to estimated intrinsics, showing only minor degradation and demonstrating its suitability for deployment in unconstrained settings.

**Label noise in KITTI.** Since 3D bounding box annotations depend on the sparsity and distribution of LiDAR points, obtaining labels as precise as their 2D counterparts is inherently challenging. As shown in Fig. 6 in the main paper and Fig. 15, both 3D-MOOD (Yang et al., 2025) and PLOT occasionally produce detections that better align with visual evidence than the provided ground-truth, where some objects are missing or incompletely annotated. This not only supports our earlier finding that PLOT can serve as a reliable substitute for manual 3D labeling, but also highlights its potential for facilitating future dataset creation by bootstrapping new benchmarks with reduced human annotation effort. Although KITTI remains a valuable benchmark, certain limitations in its labels can lead to an underestimation of the method's performance.

# E   FAILURE MODE ANALYSIS

While the quantitative and qualitative results in the main paper demonstrate the robustness of our method, it does not perform flawlessly in every scenario. This section analyzes representative failure cases, as well as situations where the proposed approach cannot theoretically provide substantial benefits. As summarized in Fig. 11, our failure modes fall into three major categories.

**Case 1: Limited relative motion.** Our approach captures and fuses shape variations through relative motion between the camera and objects. However, when the relative pose remains nearly constant-for instance, due to minimal ego-motion or objects moving steadily at a distance—the method offers limited advantages, as shown in Fig. 11-(1). In such cases, long-range objects provide insufficient evidence for label refinement because 2D detections remain sparse across frames, often resulting in detection failures. This behavior is evident in Fig. 12, where the top example from Waymo shows a case in which a subsequent right turn of the camera causes the object to move out of view, leaving the occlusion unresolved, and the bottom example depicts an object persistently hidden by occlusion, preventing reliable observations.

**Case 2: Long-range depth degradation.** As with other open-set methods or monocular depth estimators, depth noise grows with distance from the camera, leading to degraded accuracy in center

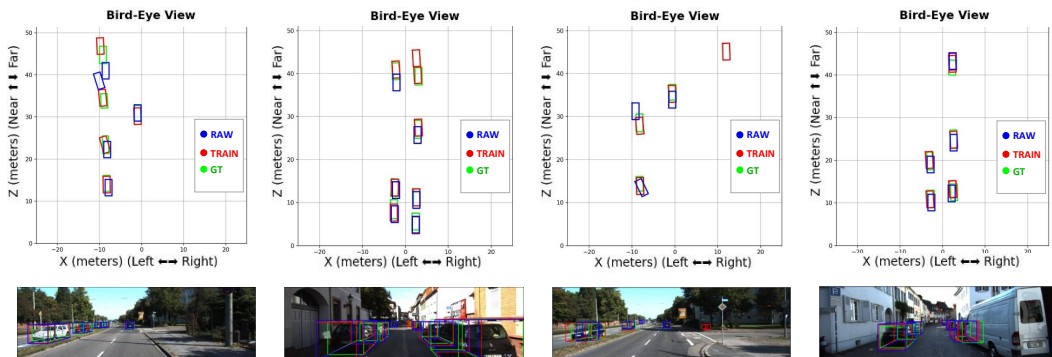

Figure 13: **BEV comparison between the detection results of raw pseudo-labels and the detector trained with it, on KITTI (Geiger et al., 2012).**

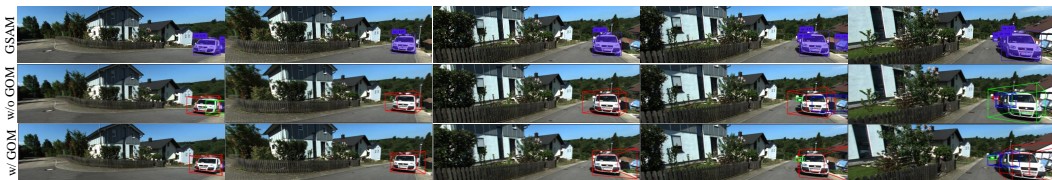

Figure 14: **Qualitative comparisons with and without global object memory, on KITTI-360 (Liao et al., 2022).** Zoom for the best view.

estimation. As shown in Fig. 11-(2), even though our method generally produces accurate labels, the estimated center gradually shifts as objects recede. While such issues may be alleviated as monocular depth models improve, they remain a practical bottleneck in long-range scenarios.

**Case 3: Edge-case association errors.** Although the proposed Global Object Memory (GOM) substantially improves object association compared to conventional approaches (see Fig. 8), it cannot resolve all cases. Fig. 11-(3) illustrates such an edge case: objects located near the boundary of the tracking window with minimal observations, or objects persistently embedded in clutter (similar to Case 1), may fail to be associated correctly. This leads to inaccurate pseudo-LiDAR fusion and, consequently, erroneous 3D bounding box estimates. These cases highlight the limits of our current association strategy and suggest that integrating stronger priors on temporal continuity or leveraging multi-view geometric cues could further enhance robustness.

# F ADDITIONAL QUALITATIVE RESULTS

**Raw pseudo-labels vs. trained labels.** As a complement to the standalone detector ablation in Tab. 5 of the main paper, Fig. 13 compares ground truth, raw pseudo-labels, and the results of MonoDETR (Zhang et al., 2023) trained on them in BEV. While the raw pseudo-labels already provide reasonable object detections, depth noise often shifts the estimated centers. Training mitigates these errors by regularizing noisy depth estimates, leading to more accurate and stable center predictions. Nevertheless, our estimated pseudo-labels demonstrate reasonable localization performance even without training, highlighting their potential as a standalone detector.

**Necessity of global object memory.** As a complement to the global object memory ablation in Tab. 5 of the main paper, Fig.14 illustrates per-frame results over five consecutive frames, showing (top) per-frame predicted masks from GSAM (Ren et al., 2024), (middle) 3D bounding boxes estimated without global object memory (GOM), and (bottom) 3D bounding boxes estimated with GOM. With GOM, the predicted boxes become progressively more accurate and consistent in size and orientation as camera motion provides additional observations. In contrast, without GOM, incorrect associations give rise to erroneous boxes and ID switches. The importance of consistent association for stable shape fusion is further supported by Fig.9, as discussed in Appendix C.

**Qualitative results on KITTI (Geiger et al., 2012).**   Fig. 15 presents qualitative comparisons on KITTI against ground truth, a supervised open-set detector (3D-MOOD) (Yang et al., 2025), and a pseudo-labeling method (MonoSOWA) Skvrna & Neumann (2025). Notably, PLOT produces 3D boxes that are visually on par with 3D-MOOD, despite the latter being trained with full 3D supervision. In contrast, MonoSOWA frequently generates oversized boxes, reflecting the limitations of its LLM-driven (Achiam et al., 2023) shape priors. These results show that PLOT achieves high reliability without 3D supervision while also exposing the shortcomings of LLM-driven geometric reasoning.

**Qualitative results on Waymo-Open (Sun et al., 2020).**   Fig. 16 presents qualitative results across diverse conditions: a daytime scene (left), a challenging environment with rain or light saturation (middle), and a nighttime scene (right). Ground-truth annotations are shown in green boxes, while our predicted results are shown in red. In both the daytime and challenging-condition scenes, our method produces detections that closely align with the ground truth, even under heavy clutter and adverse conditions. Remarkably, in the nighttime scene, our trajectory-guided labeling captures additional object instances that are missing in the ground-truth labels, demonstrating the robustness of our approach under low-visibility settings. These qualitative results highlight the practical potential of our method for monocular 3D object detection in real-world scenarios.

**In-the-wild videos (Contributors, 2025; Milan et al., 2016).**   In addition to the cross-domain results shown in Fig. 1, Fig. 2, and Fig. 7 of the main paper, we compare our raw pseudo-labels with those from the open-set single-image pseudo-labeler (OVM3D-Det) (Huang et al., 2024b) and the open-set 3D object detector (3D-MOOD) (Yang et al., 2025), in Fig. 17. OVM3D-Det, relying on single-image and shape priors, consistently produces inaccurate box sizes and center estimates, while 3D-MOOD suffers from reduced recall in novel domains. In contrast, our method consistently provides reasonable estimates across diverse conditions, including challenging cases such as phone-captured vertical videos (last row). To further demonstrate spatial consistency in in-the-wild videos, Fig. 18 provides both 2D projections and BEV visualizations for a scene from MOT17 (Milan et al., 2016). From the BEV view, objects are shown to lie on a consistent ground plane with well-aligned relative positions, offering clearer evidence of the geometric coherence provided by our pipeline, whereas 3D-MOOD mainly struggles with relative spatial positioning across frames.

# G   USAGE OF LARGE LANGUAGE MODELS

Large language models (LLMs) were used only for writing assistance, such as polishing text and grammar checking, and did not contribute to the technical contents of this work. In particular, they were not involved in the retrieval or discovery of related literature, research ideation, analysis, or any other part of the scientific process. Our method itself does not involve direct LLM usage. However, for evaluating a baseline (OVM3D-Det) (Huang et al., 2024b) that leverages LLM-based shape priors (Achiam et al., 2023) in the in-the-wild video setting, we prompted an LLM to estimate approximate object sizes, which were then used as input priors for that baseline.

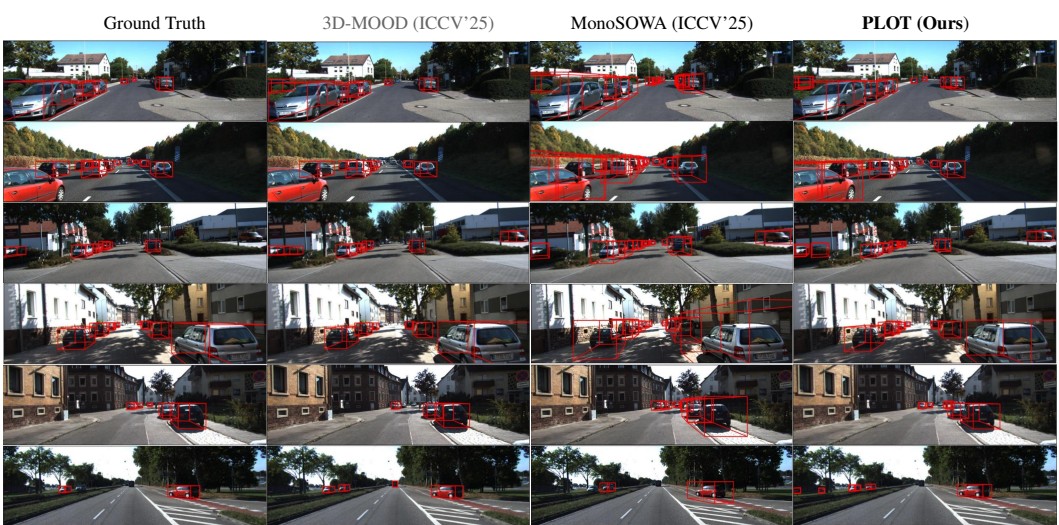

Figure 15: **Qualitative comparisons on KITTI (Geiger et al., 2012).** The supervised method-3D-MOOD (Yang et al., 2025) is shown in gray.

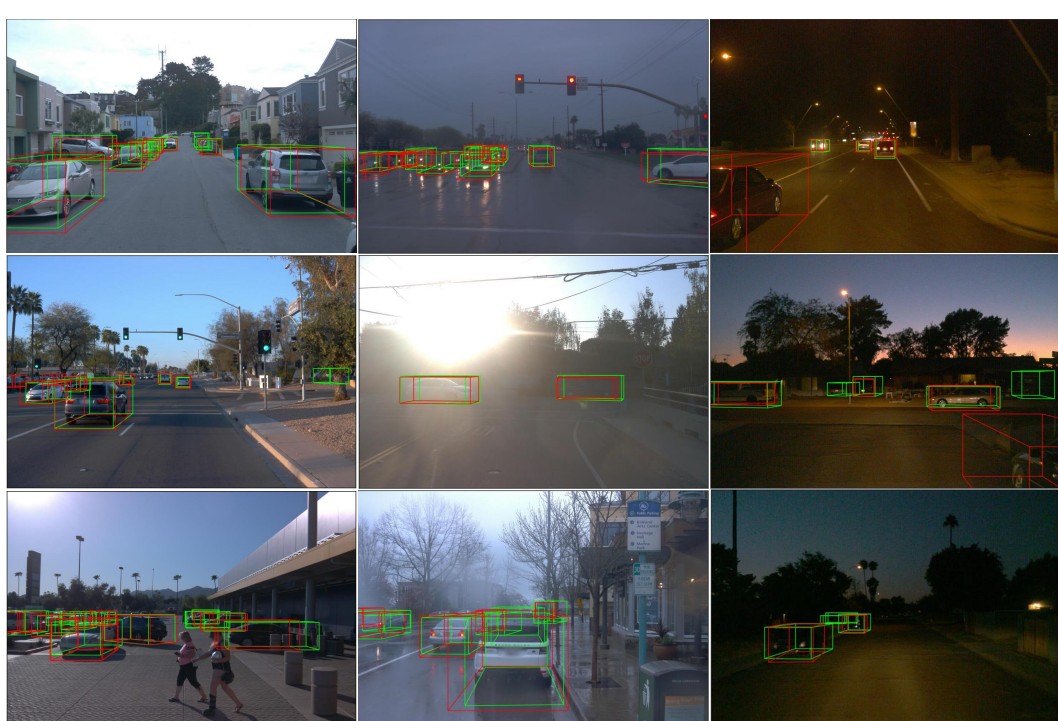

Figure 16: **Qualitative results on Waymo (Sun et al., 2020) under challenging conditions.** Left: standard driving scenes. Middle: scenes with challenging environmental conditions (e.g. raining, light-saturation). Right: night scenes with limited illumination. Our trajectory-guided labeling improves temporal consistency and yields reliable pseudo-labels even in such adverse scenarios. Ground truth boxes are shown in green and predicted boxes in red.

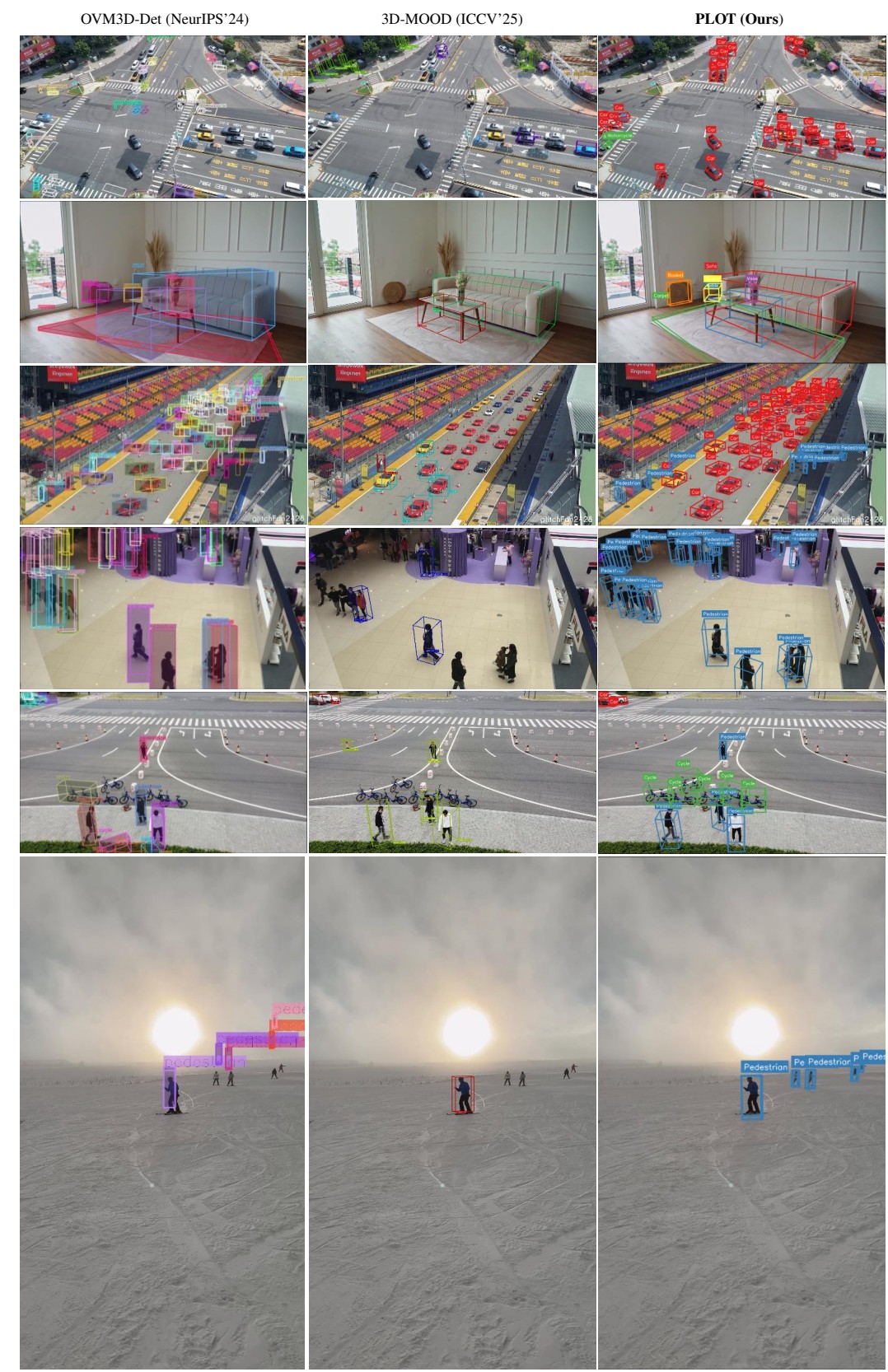

Figure 17: **Additional qualitative results on in-the-wild videos.** Prompts with same object classes are used for all methods.

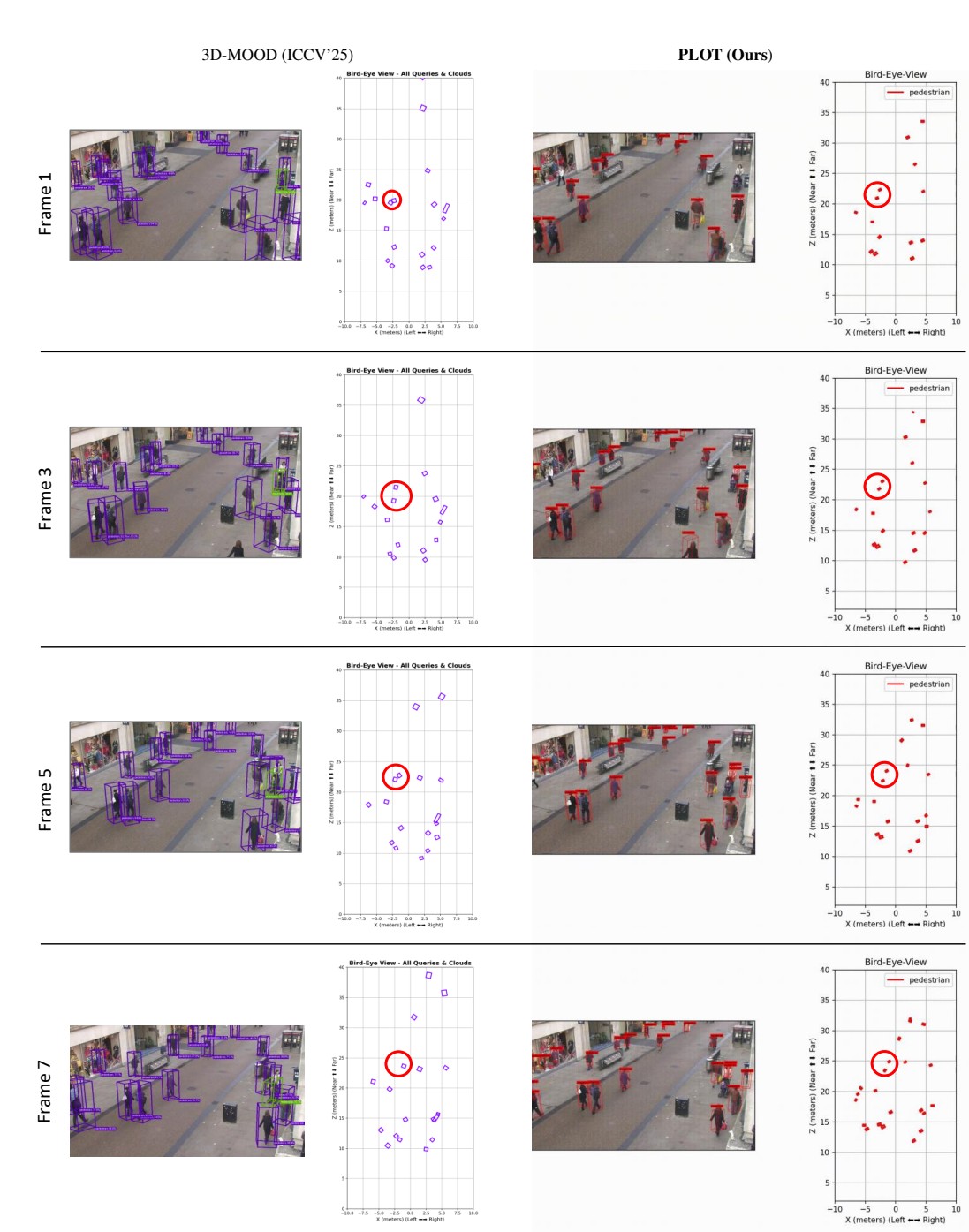

Figure 18: **Additional qualitative comparisons on MOT17 (Milan et al., 2016) sequence visualized on BEV.** The red dotted circles highlight regions where our method maintains temporally consistent ground-plane geometry and stable 3D box estimates, in contrast to the spatial drift observed in 3D-MOOD (Yang et al., 2025).

