# OpenReview forum: "PLOT: Pseudo-Labeling via Object Tracking for Monocular 3D Object Detection"
_ICLR.cc/2026/Conference — Submitted to ICLR 2026_

### Official Review · Reviewer_pYBp · 2025-10-30

**Soundness:** 4
**Presentation:** 3
**Contribution:** 3
**Rating:** 6
**Confidence:** 4

**Summary:**

This paper proposes a novel annotation-driven strategy to enhance 3D object detection, introducing a semi-automatic labeling technique that aims to reduce manual annotation costs while maintaining detection accuracy. The method combines geometric priors with weakly supervised signals to generate pseudo labels, which are then applied to 3D detection tasks on the KITTI dataset. Experimental results show measurable improvements compared to several baselines.

**Strengths:**

1. The proposed labeling strategy provides a creative solution to the high-cost bottleneck of manual 3D annotations, which is an important and underexplored aspect of 3D detection research.
2. The paper is well-motivated and clearly connects the challenges in 3D annotation with detection performance, demonstrating technical feasibility through the KITTI experiments.

**Weaknesses:**

1. The method is only validated on KITTI, a relatively small and well-studied dataset.To establish broader applicability, evaluation on additional datasets such as nuScenes, Waymo.
2. It is not explicitly clarified whether the same 3D detection or configuration is used across the proposed and compared methods. Since the labeling technique is applied to detection task, any difference in detection could confound the comparison results. A clear statement on this consistency is necessary to validate fairness.

**Questions:**

Please see the weaknesses.

---

> ### Author Response · Authors · 2025-11-17
> **Response to Reviewer pYBp**
>
> We thank the reviewer for recognizing the importance of our proposed solution in 3D detection research. Below, we address the reviewer's questions in details.
>
> ---
>
> ### [W1] Evaluation only on KITTI?
>
> We agree that broader validation is important. In addition to KITTI, **we also evaluate our method on the larger KITTI-360 and Waymo-open datasets (see Tables 2 and 3)**. Our approach consistently outperforms previous open-set detection, weakly supervised, and pseudo-labeling methods across these benchmarks, demonstrating strong generalization beyond the standard KITTI setup. Complementary qualitative results are also provided in the main paper (Figs. 5 and 6) and in the appendix (Figs. 15 and 16).
>
> ---
>
> ### [W2] Configuration of comparison methods?
>
> We ensure a fair comparison with previous pseudo-labeling methods by using consistent detection and depth configurations wherever possible. Specifically, **we follow OVM3D-Det**, which uses Grounded-SAM as the open-vocabulary detector and UniDepth as the depth estimator, while MonoSOWA uses a COCO-trained MViTv2 detector and Metric3Dv2 for depth; both depth estimators show similar performance based on reported results. All pseudo-labeling methods are trained with MonoDETR using default hyperparameters. We clarified these experimental details in the updated manuscript (Appendix.A) to explicitly highlight consistency across methods.
>
> ---

---

> ### Author Response · Authors · 2025-11-28
> **Follow-up**
>
> Dear Reviewer pYBp
>
> We hope this message finds you well. We are writing to kindly follow up regarding the responses we provided to your comments. We have clarified where each of the relevant results can be found in the revised manuscript and appendix, and we want to ensure that our explanations adequately address the concerns you raised.
>
> If any part of our clarification remains unclear or if you would like further details, we would be very happy to provide additional explanation. As the rebuttal deadline is approaching, we would greatly appreciate any confirmation or further feedback at your convenience.
>
> Thank you again for your time and thoughtful review.
>
> Best regards,
> Authors

---

### Official Review · Reviewer_9QpT · 2025-10-31

**Soundness:** 3
**Presentation:** 2
**Contribution:** 3
**Rating:** 4
**Confidence:** 3

**Summary:**

This paper introduces PLOT (Pseudo-Labeling via Object Tracking), a novel framework for generating 3D object detection annotations from monocular videos. The core contribution is a training-free pipeline that does not require auxiliary sensors (like LiDAR) or model retraining. This addresses the critical challenges of 3D annotation scarcity and the poor generalization of existing models to "in-the-wild" scenarios.

**Strengths:**

- The primary strength is the training-free nature of the pseudo-labeling framework. This frees the method from reliance on specific datasets or sensors (e.g., LiDAR or known camera poses), demonstrating strong generalization to "in-the-wild" videos, which is a major limitation of current M3OD methods.
- The method's "trajectory-guided shape fusion" aggregates information from multiple frames to build a "completed pseudo-LiDAR". This effectively addresses occlusion, truncation, and sparse point cloud issues that plague single-frame methods, leading to more accurate attribute estimation. The ablation in Table 4 confirms that using more frames (e.g., 20) for fusion significantly improves AP.
- The pseudo-labels generated by PLOT lead to excellent performance when used for training.

**Weaknesses:**

- The method estimates orientation for dynamic objects based on their motion. However, for static objects, it falls back to using PCA on the fused pseudo-LiDAR. While the authors claim the denser fused cloud makes PCA more robust, this remains a strong heuristic. This approach may fail for objects with near-symmetrical shapes (like vans) or where the principal axis of variance does not align with the object's heading.
- While the framework is presented as "training-free," its performance is fundamentally dependent on several complex, pre-trained foundation models, including a 2D detector (GSAM), a dense point tracker (AllTracker), and a monocular depth estimator (UniDepth/MoGe2). A critical issue is the lack of discussion on the temporal consistency of these depth estimates from a monocular estimator. How does PLOT mitigate potential temporal flickering or scaling inconsistencies in the depth predictions from frame to frame?

**Questions:**

- Table 4 analyzes the computation time for shape fusion. However, to assess practical utility, what is the end-to-end latency of the entire PLOT pipeline (from video input to pseudo-label output)? This should include the full overhead of running GSAM, AllTracker, and UniDepth.
- The paper uses an advanced dense point tracker (AllTracker), yet also introduces GOM to resolve tracking failures like ID switches. To what extent is GOM correcting for AllTracker's own tracking errors versus primarily handling detection dropouts from the 2D detector (GSAM)? The line between these two error sources seems blurry.
- Camera ego-motion is estimated via Procrustes alignment (Eq. 4) on background points lifted to 3D by the depth estimator. How robust is this alignment to noise from the monocular depth predictions, especially non-uniform noise? How does depth error propagate to the camera motion estimate and, subsequently, to the quality of the final shape fusion?

---

> ### Author Response · Authors · 2025-11-17
> **Response to Reviewer 9QpT (1/2)**
>
> We sincerely appreciate the reviewer's suggestions and constructive feedback. We are grateful to address them in details.
>
> ---
>
> ### [W1] Concern with PCA for static objects.
>
> We appreciate the reviewer’s observation, and we agree that PCA can be ambiguous for near-symmetric objects or in cases where the principal axis of variance does not align with the true heading. In our setting, however, the static-object orientation problem is fundamentally ill-posed under single-view observations, and PCA has remained a commonly adopted fallback in recent practice for handling such cases. **Our aim is not to fully resolve this inherent ambiguity but to provide a stable and practical estimate within a retraining-free pipeline. PCA on a denser fused cloud offers reliable orientation in the majority of driving scenarios, while dynamic objects benefit from our motion-based estimation.** Incorporating an additional trained orientation estimator with semantic cues could further reduce ambiguity, but this lies beyond the scope of the present work. We thank the reviewer for raising this point and will clarify these limitations more explicitly in the revised draft.
>
> ---
>
> ### [W2] Addressing temporal flickering of depth predictions.
>
> We appreciate the reviewer’s insightful comment, and we acknowledge that the term 'training-free' may cause confusion; accordingly, we have revised the terminology in the paper to 'without retraining', since our method uses pre-trained foundation models. Regarding potential temporal or scale inconsistencies in monocular depth predictions, **our experiments show that UniDepth and MoGe2 produce stable inter-frame scales in large-scale, non–object-centric scenes**. We demonstrate this stability through both the supplemental video and the newly added frame-wise BEV visualization in Fig. 18 (e.g., the two pedestrians at the lower-left corner of frame 1), which exhibit minimal flickering or scale drift across consecutive frames. **Our motion estimation uses SE(3) alignment during point registration, and can also be extended to Sim(3) if larger temporal scale variations are observed.** We have also updated the Section 4.2 to clarify these points accordingly.
>
> ---
>
> ### [Q1] Computation time for whole pipeline?
>
> **The latency test reported on Table. 4 is for the whole PLOT pipeline** including priors generation.
> We apologize for the confusion, and for clarity we provide the runtime for each component as follows:
>
> Component | Latency (s)
> --- | ---
> GSAM | 0.30
> UniDepth | 0.21
> AllTracker | 0.24
> Pseudo-labeling | 1.70
> All | 2.45
>
> ---
>
> ### [Q2] Extent of GOM?
>
> We appreciate the reviewer for raising this subtle but important point, as clarifying the division of roles between AllTracker and GOM indeed helps convey the intended behavior of our system. AllTracker provides dense and reliable correspondences, but under strong occlusion or missing detections, its tracking may break, splitting a single object trajectory into two disconnected segments—conditions that closely mirror detector-induced dropouts. GOM is designed to address these issues in two ways: **1) its primary role is to restore continuity when the detector produces missed or duplicated detections**, and **2) it reconnects trajectory pieces that become separated when occlusion reduces point visibility below AllTracker’s threshold.** In both cases, GOM does not modify AllTracker’s tracking process, but instead **re-establishes continuity where observations temporarily vanish.** We clarified this distinction more explicitly in Section 4.3 in the revised manuscript.

---

> > ### Author Response · Authors · 2025-11-19
> > **Response to Reviewer 9QpT (2/2)**
> >
> > ---
> >
> > ### [Q3] Impact of depth noise on camera motion and shape fusion?
> >
> > We appreciate the reviewer’s thoughtful feedback regarding the stability of our ego-motion and shape fusion under depth noise and temporal scale inconsistencies. In response, we have added a dedicated ablation study here and also in the revised manuscript (Table.6 and the subsection “Robustness to depth noise and temporal scale inconsistencies”).
> > This analysis injects controlled range-dependent noise, spatially non-uniform noise and temporal scale perturbations into the UniDepth predictions on a sampled Waymo-Open sequence, and evaluates their effect on depth accuracy, camera motion estimation, and final 3D labeling quality against ground truth.
> >
> > Alignment | Range Noise | Patch Noise | Temporal Jitter | Depth RMSE ↓ | Pose ATE ↓ | AP3D@0.3 Easy ↑ | AP3D@0.3 Hard ↑ | APBEV@0.3 Easy ↑ | APBEV@0.3 Hard ↑ |
> > --- | ----------- | ----------- | --------------- | ------------ | ---------- | --------------- | --------------- | ---------------- | ----------------
> > SE(3) | x | x | x | **3.72**     | **0.204**  | **20.89**       | **14.38**       | **27.85**        | **20.20**
> > SE(3) | 0.04 | 0.3 | x    | 3.90 (+0.18) | 0.205     | 20.30 (-0.59) | 14.00 (-0.38) | 28.80 (+0.95) | 19.73 (-0.47)
> > SE(3) | 0.08 | 0.3 | x    | 4.34 (+0.62) | **0.204** | 19.48 (-1.41) | 13.45 (-0.93) | 26.63 (-1.22) | 19.35 (-0.85)
> > SE(3) | 0.08 | 0.6 | x    | 4.38 (+0.66) | 0.206     | 19.65 (-1.24) | 13.39 (-0.99) | 26.94 (-0.91) | 19.39 (-0.81)
> > SE(3) | x    | x   | 0.02 | 3.78 (+0.06) | 0.205     | 17.50 (-3.39) | 11.96 (-2.42) | 25.84 (-2.01) | 17.43 (-2.77)
> > SE(3) | x    | x   | 0.04 | 3.82 (+0.10) | **0.204** | 14.14 (-6.75) | 9.49 (-4.89)  | 20.29 (-7.56) | 14.54 (-5.66)
> > Sim(3) | x    | x  | 0.02 | 3.78 (+0.06) | **0.204** | 17.69 (-3.20) | 13.39 (-0.99) | 25.96 (-1.89) | 19.01 (-1.19)
> > Sim(3) | x    | x  | 0.04 | 3.81 (+0.09) | **0.204** | 14.99 (-5.90) | 11.23 (-3.15) | 22.55 (-5.30) | 16.36 (-3.84)
> >
> > The results show that camera motion estimation remains stable due to the larger number of reliable background correspondences, and that spatial depth noise leads only to mild degradation in fused geometry and 3D attributes. In the case of non-uniform temporal scale noise, it introduces significantly larger position drift of objects compared to the depth accuracy of whole image. While this issue is inherent to any pseudo-labeling pipeline relying on monocular depth-since evaluation is performed against absolute metric ground truth. It can be mitigated by using Sim(3) alignment, as reflected by the improvements in the last two rows of the table. Empirically, modern monocular depth estimators exhibit only slight temporal scale inconsistencies in driving scenes (main benchmarks) and general videos with large-scale scenes (attached supplemental videos and updated Fig.18), and those do not significantly affect PLOT's fused geometry.
> >
> > Finally, we emphasize that monocular metric depth remains essential for scalable in-the-wild labeling, and future advances in monocular depth estimation are expected to further improve the robustness and quality of PLOT. We appreciate the reviewer's constructive feedback and will continue to refine this aspect in future work.

---

> > > ### Comment · Reviewer_9QpT · 2025-11-26
> > >
> > > The authors have resolved my concerns, so I'm willing to raise my rating.

---

> > > > ### Author Response · Authors · 2025-11-28
> > > >
> > > > We sincerely appreciate the reviewer’s thoughtful reassessment and for raising the rating. If there are any further questions or concerns regarding our work, we would be more than happy to address them.

---

### Official Review · Reviewer_7swF · 2025-11-01

**Soundness:** 3
**Presentation:** 4
**Contribution:** 3
**Rating:** 6
**Confidence:** 3

**Summary:**

This paper proposes a method for pseudo-labeling 3D bounding boxes around objects of interest from in-the-wild videos. They propose a pipeline where they track points on objects and on the background to derive both camera motion and object motion. Then, they use an off-the-shelf metric depth estimator to lift these points into 3D, and aggregate points of objects of interest to derive a 3D bounding box around them, with the heading angle derived from the object motion in world frame. Using their pseudo-labels, they demonstrate strong performance on KITTI and Waymo datasets, and also present an interesting demo of pseudo-labels on in-the-wild video.

**Strengths:**

- The proposed pipeline is intuitive and reasonable, drawing from recent improvements in 2D tracking and effectively using them to derive 3D motion and labels.
- While a straightforward use of PA, the work derives reasonable estimates for camera and object motion.
- The improvement from training on the pseudo-labels is good to see in Figure 13.
- The proposed method demonstrates strong results compared to prior work in KITTI and Waymo.
- The ablation study in Tables 4 and 5 are reasonable, showing improvement with more frames.

**Weaknesses:**

- While the pipeline is reasonable, this reviewer wonders if it's necessary to derive camera motion in such a manner. For instance, if we put the entire sequence into COLMAP, or perhaps some SLAM method, could it recover relative camera poses similarly reasonably? Such poses can directly be lifted to metric scale using the same metric depth estimator. Given advancements in the field, MonSt3R-based SLAM and VGGT must also be considered as possible system-level competitors as well, as they directly output 3D points, which one can derive 3D boxes from.
- This paper seems to derive tracklets by propagating tracked masks from one frame to the next using an off-the-shelf point tracker. New masks are matched via hungarian matching and the authors also present GOM, which appears to be a tracklet rebirth mechanism. This is a reasonable, performant pipeline, but it's difficult to say it is this paper's contribution as I believe these are common practices in object tracking frameworks.
- Building on the previous point, such pipelines often prove to be brittle to unexpected motion and perhaps have difficulty with long-term consistency. I would like to ask if an off-the-shelf model like SAM2, which directly outputs mask tracklets could be a one-touch alternative that performs well. To the best of my understanding, the input & output is the same: input RGB video, output tracked masks.
- While it is good to see visualizations of boxes on in-the-wild videos, it is difficult to truly assess the quality of 3D from the 2D projection (since the 3D was derived from that 2D frame, it's easier for it to look aligned). Some visualizations of point clouds or boxes in another view could prove important: for instance, visualizing the crossroad image at the top of Figure 17 from a road-level view to assess if all objects are on a consistent plane.
- Is the performance improvement in Table 4 with more frames purely due to a depth of a single frame being unable to give the full _extent_ of an object? If so, would reasonable priors on the size of a car in general serve to improve performance of a single-frame baseline?

**Questions:**

I would appreciate it if the authors address my concerns regarding simpler alternatives, as outlined in the first few points in the Weaknesses section. The authors provide a reasonable and performant pipeline, but it's not yet clear to this reviewer that all such components are necessary, especially in light of advancements in video models. As such, at this stage, I recommend a 6.

---

> ### Author Response · Authors · 2025-11-17
> **Response to Reviewer 7swF (1/2)**
>
> We sincerely thank the reviewer for giving time and effort to review the paper. We want to address the reviewer's concern in details.
>
> ---
>
> ### [W1] 3D annotations from reconstruction-based methods?
>
> We appreciate the reviewer’s suggestion and acknowledge that recent reconstruction-centric systems (e.g., VGGT [1], MonSt3R [2]) are **strong baselines for global scene recovery**. However, these **SLAM-style pipelines either ignore dynamic objects (e.g.,VGGT [1], MASt3R [3]) or handle only a very restricted number of moving entities (e.g., MonSt3R [2], CUT3R [4]),** whereas our setting explicitly leverages object motion as a constructive multi-view cue for completing object shape. **Our focus is object-centric 3D perception rather than full-scene reconstruction**, and this distinction leads to different design priorities and failure modes. Moreover, reconstruction-based pipelines still require an external Sim(3) alignment to obtain metric-scale poses, while our static-background correspondence approach yields metric camera motion directly and with minimal complexity.
>
> In addition, the purpose of our ego-motion estimation is not global pose-graph optimization but ensuring that per-frame object poses are expressed in a consistent world coordinate frame for reliable orientation reasoning. For this object-centric goal, **our tracking-based alignment provides a lightweight and robust alternative** to reconstructed-scene SLAM modules, avoiding additional dependencies while remaining well-suited to dynamic environments. We thank the reviewer for raising this point and clarified these distinctions explicitly in the Introduction section of revised draft.
>
> [1] Wang, Jianyuan, et al. "Vggt: Visual geometry grounded transformer." CVPR, 2025.
>
> [2] Zhang, Junyi, et al. "Monst3r: A simple approach for estimating geometry in the presence of motion." ICLR, 2025.
>
> [3] Leroy, Vincent, Yohann Cabon, and Jérôme Revaud. "Grounding image matching in 3d with mast3r." ECCV, 2024.
>
> [4] Wang, Qianqian, et al. "Continuous 3d perception model with persistent state." CVPR, 2025.
>
> ---
>
> ### [W2] Concern with tracklet management system.
>
> We agree that initialization and re-matching are common components in tracking frameworks, but **our contribution lies in introducing dense point tracking into a 3D object detection and labeling pipeline, where temporally grounded 3D annotations must be produced without retraining the tracker.** In this setting, object visibility naturally changes due to motion and occlusion, which can lead to fragmented trajectories even when using a strong tracker. GOM is designed to reconnect these fragmented segments so that object tracks remain coherent over long temporal ranges. **While the individual elements resemble familiar tracking practices, their integration is essential for producing reliable, temporally consistent 3D annotations in a re-training-free pipeline.**
>
> ---
>
> ### [W3] SAM2 as a one-touch alternative for mask tracklets?
>
> We appreciate the reviewer’s suggestion and agree that SAM2 [5] can directly produce video masks. However, **SAM2 relies on prompt-based initialization and does not provide a mechanism to autonomously incorporate newly appearing objects into the tracking state**, which frequently occurs in driving scenes. In practice, variants such as Grounded-SAM2 extend SAM2 with simple mask matching, but this approach often breaks under occlusions-leading to fragmented tracks or ID switches-and is also sensitive to large ego-motion. In contrast, our method combines a long-term dense point tracker with GOM to robustly manage new, persistent, and departing objects. This design yields substantially improved temporal continuity and robustness compared to the naive mask-matching strategies in Grounded-SAM2 and prior pseudo-labeling frameworks such as MonoSOWA.
>
> [5] Ravi, Nikhila, et al. "Sam 2: Segment anything in images and videos." arXiv preprint arXiv:2408.00714 (2024).
>
> ---

---

> > ### Author Response · Authors · 2025-11-18
> > **Response to Reviewer 7swF (2/2)**
> >
> > ### [W4] Visualizations from another view?
> >
> > We appreciate the reviewer’s suggestion regarding more informative 3D visualizations. To complement the 2D projections, **we have added Fig. 18, which presents both the image-plane and BEV projections for in-the-wild examples** from MOT17. While BEV does not encode full 3D height, it serves as a strong proxy for assessing ground-plane consistency, as misaligned elevations would appear as spatial drift in the bird’s-eye view. This visualization clearly shows that **all objects lie on a consistent ground plane and maintain well-aligned spatial relationships across views and temporal frames**, compared to 3D-MOOD. We have also clarified this point in the “In-the-wild Videos” paragraph of Appendix Section F.
> >
> > ---
> >
> > ### [W5] Performance improvement in Table. 4?
> >
> > We appreciate the reviewer’s insightful question. The improvements in Table 4 arise from **two complementary benefits of multi-frame aggregation: 1) completing the spatial extent of objects using mutually informative observations across frames, and 2) recovering objects that may be missing in individual frames**, which establishes essential temporal consistency. Prior-based methods such as OVM3D-Det and MonoSOWA already incorporate size or template priors, but these approaches must still align such priors under partial and viewpoint-specific observations (e.g., when only the front or side of a vehicle is partially visible), which limits their reliability in single-frame settings. Despite relying solely on geometric cues accumulated across frames—without priors or retraining—our simple and flexible pipeline already surpasses these prior-based approaches, as reported in Table 1, 2, and 3.
> >
> > ---

---

> ### Author Response · Authors · 2025-11-28
> **Follow-up**
>
> Dear Reviewer 7swF
>
> We hope this message finds you well. We are writing to kindly follow up on the concerns you previously raised. We have provided detailed responses to each point to the best of our ability, and we would be happy to offer additional clarification if any part of our reply remains insufficient.
>
> Since the rebuttal deadline is approaching, we would greatly appreciate any confirmation or additional feedback you may have regarding our responses. Your guidance will help us ensure that we address your concerns as thoroughly and accurately as possible.
>
> Thank you again for your time and thoughtful review.
>
> Best regards,
> Authors

---

### Official Review · Reviewer_y8oj · 2025-11-01

**Soundness:** 3
**Presentation:** 3
**Contribution:** 3
**Rating:** 4
**Confidence:** 3

**Summary:**

This paper presents SafeScale, a novel and scalable framework for generating synthetic driving data to improve the safety and robustness of autonomous driving planners. The core problem addressed is the difficulty of collecting real-world data for rare and dangerous "corner cases" that often cause modern data-driven planners to fail.

**Strengths:**

Strengths
1.	The paper tackles a critical and high-impact problem in autonomous driving: improving safety by addressing corner cases. The primary contribution—a clear, empirical demonstration that scaling targeted synthetic data can directly and significantly improve a planner's performance in the real world.
2.	The SafeScale framework is well-designed and technically sound. The idea of decomposing real scenes into modular asset libraries (backgrounds, appearances, behaviors) is a powerful concept that enables both scalability and fine-grained control.
3.	The experimental evaluation is exemplary. The main result, presented in Figure 4, provides convincing evidence of a "synthetic data scaling law," which is the paper's central claim. The ablation studies are insightful and strongly support

**Weaknesses:**

Weaknesses:

1.	The contribution can be characterized as a sophisticated and highly successful data engineering framework. Its primary novelty lies in the clever integration of existing components and the powerful empirical demonstration of the scaling law, rather than in a fundamental algorithmic advance. The method for generating corner cases is 'reactive'—it relies on first analyzing the specific failure modes of a baseline planner (DiffusionDrive) on a specific benchmark (NAVSIM). A more scientifically profound direction, which the paper does not explore, is how to proactively and universally model and generate corner cases. For instance, could one develop a general model that learns the underlying distribution of safe driving data and then synthesizes challenging out-of-distribution scenarios in a principled way, without being tied to the failures of one specific planner? This would elevate the contribution from a highly effective, bespoke data augmentation solution to a more fundamental model of driving risk.
2.	The traffic participant behavior library is built from trajectories observed in the NAVSIM dataset. While the framework can create novel scenarios by placing these behaviors in new contexts, it is fundamentally limited by the vocabulary of behaviors present in the source data. The paper would be strengthened by a discussion of this limitation. Can this method generate truly novel, out-of-distribution behaviors, or is it primarily a powerful recombination engine?
3.	The paper states that the asset extraction pipeline is highly scalable as it relies on sensor data and 3D annotations. However, the acquisition of clean, large-scale 3D annotations is a known bottleneck and a significant cost factor for the entire industry. A brief discussion on the sensitivity of the SafeScale pipeline to the quality and scale of these initial annotations would be welcome. For instance, how do sparse or noisy annotations affect the quality of the generated assets and the final planner performance?
4.	The use of a generative model for view synthesis is a key strength, but these models are not perfect. They can introduce subtle artifacts, temporal inconsistencies, or a lack of physical realism (e.g., incorrect shadows, reflections). The paper does not discuss the potential sim-to-real gap of this rendering stage. While the end-to-end results prove the data is highly effective, a qualitative analysis or discussion of the generative renderer's failure modes would add nuance and provide a more complete picture

**Questions:**

na

---

> ### Author Response · Authors · 2025-11-12
> **Possible Misalignment Between Review Comments and Our Submission**
>
> We appreciate your time and feedback. We noticed that the review discusses elements that do not appear in our submission, suggesting it may have been meant for a different paper. We would be grateful if you could verify this and provide clarification, which would help us respond appropriately within the rebuttal timeframe.

---

> > ### Comment · Reviewer_y8oj · 2025-11-26
> > **Fix the wrong review**
> >
> > I have revised my review comments, and recommend a borderline accept decision as the manuscript holds sufficient scientific value.

---

> > > ### Author Response · Authors · 2025-11-29
> > > **Corrected Review by Reviewer y8oj**
> > >
> > > We would like to briefly clarify the situation regarding Reviewer y8oj’s review. **The initial review attached to our submission was mistakenly written for a different paper.** Before the system closure, the reviewer provided a corrected review, and our rebuttal was written entirely in response to this corrected version. However, due to the system disruption, the updated review no longer appears in the interface.
> > >
> > > For transparency, we include below the corrected review that Reviewer y8oj submitted prior to the unfortunate incident. This version can also be verified through the reviewer’s revision history.
> > >
> > > ---
> > >
> > > ### Corrected Review by Reviewer y8oj
> > >
> > > #### Summary
> > > The paper introduces PLOT (Pseudo-Labels via Object Tracking), a training-free framework for generating 3D object detection annotations from monocular videos. PLOT focus on two main limitation of existing methods in monocular 3D object detection, i.e., the scarcity and limited diversity of 3D annotations, as well as the inherent ambiguity of single-image geometry. To address these challenges, PLOT proposes to track object and background trajectories from monocular videos to estimate camera motion and perform object association in pose-unknown settings. Moreover, this paper introduces a Global Object Memory (GOM) that enforces video-consistent label alignment and mitigates identity switching.
> > >
> > > #### Soundness: 3
> > > #### Presentation: 3
> > > #### Contribution: 3
> > >
> > > #### Strength
> > > (1) This paper introduces a training-free pipeline for 3D pseudo-labeling, reducing reliance on expensive LiDAR or manual annotation.
> > >
> > > (2) The experimental results demonstrate the effectiveness in driving scenes, such as KITTI and KITTI-360.
> > >
> > > #### Weakness
> > > (1) Limited discussion on error propagation: while the method uses multiple foundation models, there is insufficient analysis of how errors in one stage (e.g., inaccurate depth estimation or failed tracking) affect downstream 3D attribute estimation.
> > >
> > > (2) Computational cost and runtime efficiency are not reported, which is critical for evaluating the feasibility of deploying such a multi-model pipeline in large-scale data processing pipelines.
> > >
> > > (3) The experiments presented in the manuscript are unconvincing in demonstrating that the proposed method effectively addresses the limited diversity of 3D annotations.
> > >
> > > (4) The font in the image (such as Figure 1) is too small. Moreover, it is suggested zooming in on some of the finer details in Figure 2.
> > >
> > >
> > > #### Questions: na
> > > #### Flag For Ethics Review: No ethics review needed.
> > > #### Rating: 6
> > > #### Confidence: 3

---

> ### Author Response · Authors · 2025-11-28
> **Response to Reviewer y8oj (1/2)**
>
> We sincerely appreciate the reviewer for recognizing the scientific value of our work and for dedicating time and care to providing such thoughtful feedback. We want to address the reviewer's concern in detail.
>
> ---
>
> ### [W1] Limited discussion on error propagation
> We thank the reviewer for the thoughtful and accurate observation. We agree that the original manuscript lacked a consolidated analysis of how errors originating from individual foundation model components propagate through the pipeline. In response, we now provide a dedicated study and summarize the results in the table below (also reflected in the updated Table 6 in the main paper), **explicitly illustrating how depth noise and tracking errors influence downstream 3D attribute estimation.** Our ablations introduce controlled perturbations to the estimated depth maps and different alltracker variants (less accurate but faster tiny model), allowing us to isolate their effects on camera motion estimation, fused geometry stability, and final 3D labeling quality.
>
> Alignment | Range Noise | Patch Noise | Temporal Jitter | Tiny Tracker | Depth RMSE ↓ | Pose ATE ↓ | AP3D@0.3 Easy ↑ | AP3D@0.3 Hard ↑ | APBEV@0.3 Easy ↑ | APBEV@0.3 Hard ↑ |
> --- | ----------- | ----------- | ----------- | --------------- | ------------ | ---------- | --------------- | --------------- | ---------------- | ----------------
> SE(3) | x | x | x | x | **3.72**     | **0.204**  | **20.89**       | **14.38**       | **27.85**        | **20.20**
> SE(3) | x | x | x | o | **3.72**     | **0.204**  | 20.57(-0.32)  | 14.17 (-0.21) | 27.71 (-0.14)  | 19.54 (-0.66)
> SE(3) | 0.04 | 0.3 | x | - | 3.90 (+0.18) | 0.205     | 20.30 (-0.59) | 14.00 (-0.38) | 28.80 (+0.95) | 19.73 (-0.47)
> SE(3) | 0.08 | 0.3 | x | - | 4.34 (+0.62) | **0.204** | 19.48 (-1.41) | 13.45 (-0.93) | 26.63 (-1.22) | 19.35 (-0.85)
> SE(3) | 0.08 | 0.6 | x | - | 4.38 (+0.66) | 0.206     | 19.65 (-1.24) | 13.39 (-0.99) | 26.94 (-0.91) | 19.39 (-0.81)
> SE(3) | x    | x   | 0.02 | - | 3.78 (+0.06) | 0.205     | 17.50 (-3.39) | 11.96 (-2.42) | 25.84 (-2.01) | 17.43 (-2.77)
> SE(3) | x    | x   | 0.04 | - | 3.82 (+0.10) | **0.204** | 14.14 (-6.75) | 9.49 (-4.89)  | 20.29 (-7.56) | 14.54 (-5.66)
> Sim(3) | x    | x  | 0.02 | - | 3.78 (+0.06) | **0.204** | 17.69 (-3.20) | 13.39 (-0.99) | 25.96 (-1.89) | 19.01 (-1.19)
> Sim(3) | x    | x  | 0.04 | - | 3.81 (+0.09) | **0.204** | 14.99 (-5.90) | 11.23 (-3.15) | 22.55 (-5.30) | 16.36 (-3.84)
>
> As shown in the first two rows of the table, **the overall performance drops only mildly despite the reduced tracking accuracy, indicating that global object memory effectively compensates for fragmented trajectories and helps preserve temporal consistency.** The other results show that PLOT remains stable under spatial depth noise, with more dependence on depth range noise. Under non-uniform temporal noise, PLOT can use Sim(3) alignment to mitigate temporal jitter to some extent. Importantly, we find that unlike the synthetic perturbations used in our robustness experiments, modern monocular depth estimators exhibit negligible temporal or spatial inconsistencies in large-scale scene settings, resulting in stable overall performance.
>
> We appreciate the reviewer’s valuable feedback and have incorporated this analysis into the revised manuscript to more clearly characterize error propagation within our pipeline.
>
> ---
>
> ### [W2] Computational cost and runtime efficiency report
>
> We agree that reporting computational cost is essential for evaluating the feasibility of deploying a multi-model pipeline at scale.
> In our work, **we provide runtime and hardware analyses of the entire pipeline in Table 4 and Section 5.3 of the main paper.**
> Because PLOT performs object association and motion-aware tracking, the overall computation time naturally depends on both the number of tracked objects and the temporal window used for shape fusion.
> To reflect this, Table 4 includes an ablation over different fusion frame lengths along with the corresponding per-frame runtimes.
> In this rebuttal, we also report the runtime for each component as follows:
>
> Component | Latency (s)
> --- | ---
> GSAM | 0.30
> UniDepth | 0.21
> AllTracker | 0.24
> Pseudo-labeling | 1.70
> All | 2.45

---

> ### Author Response · Authors · 2025-11-28
> **Response to Reviewer y8oj (2/2)**
>
> ### [W3] Concern regarding insufficient evidence for addressing annotation diversity
>
> We appreciate the reviewer's concern regarding the diversity of 3D annotations and the importance of demonstrating applicability beyond standard autonomous-driving benchmarks. While the absence of ground-truth 3D labels fundamentally limits quantitative evaluation in unconstrained domains, we believe our manuscript provides strong qualitative evidence that PLOT directly addresses this challenge.
>
> First, **the in-the-wild annotation results shown in Fig.1, Fig.2, and Fig.7 of the main paper, as well as the extended examples in Fig.17 and Fig.18 of the appendix and labeled videos in the supplementary material, cover a broad range of everyday scenes, weather conditions, object types, viewpoints, and camera motions that go well beyond the scope of existing 3D detection datasets.** These results consistently demonstrate that PLOT produces stable trajectories, geometrically coherent object boxes, and temporally consistent layouts, indicating that the method generalizes effectively outside structured driving environments.
>
> Second, we emphasize that the difficulty of quantitatively evaluating diversity is symptomatic of a broader limitation in current benchmarks: virtually all monocular 3D detection datasets are collected with specialized sensors (e.g., LiDAR) and therefore present only a narrow slice of real-world visual diversity. This constraint is precisely what motivates our work. PLOT provides a practical mechanism for generating reliable and scalable 3D annotations from ordinary monocular videos, **enabling 3D supervision in domains where sensor-based ground truth is unavailable or impractical.**
>
> In this sense, while numerical evaluation outside autonomous-driving settings remains an open challenge for the entire field, our qualitative results demonstrate that **PLOT meaningfully expands the diversity of sensor-denied scenarios-and thus has the potential to broaden and democratize the 3D data ecosystem.**
>
> ---
>
> ### [W4] Concern about small figure fonts and insufficient zoom on details
>
> We thank the reviewer for this helpful suggestion. We agree that the font size in Figure 1 was too small and that some details in Figure 2 were difficult to inspect. In the revised manuscript, we have increased the font sizes throughout Figure 1 for improved readability and added enhanced zoomed-in views in Figure 2 to make the finer details clearer. We appreciate the reviewer’s feedback and believe these adjustments improve the overall presentation quality.

---

### Author Response · Authors · 2025-11-29
**Clarification and Summary Regarding Our Review Discussion**

Thank you to all reviewers for their thoughtful feedback and for helping improve the quality of our paper. We incorporated additional experiments and revisions during the rebuttal, and we believe the manuscript has become substantially stronger as a result. We also want to assure the committee that we strictly followed all official ICLR guidelines throughout the rebuttal process.

We are deeply sorry for the difficulties caused by the unexpected incident this year and sincerely appreciate the ACs and PCs for taking on additional responsibilities during such a challenging period.

Before the discussion period concludes, we would like to provide a brief, consolidated clarification to ensure that our submission is evaluated fairly. **Our message concerns three points: (1) the corrected review provided by Reviewer y8oj, (2) the rating update made by Reviewer 9QpT prior to the incident, and (3) a summary of the issues we addressed during the rebuttal.**

We would also like to emphasize that we remain fully willing to respond to any further comments or clarifications.

---

### [1] Correction of Reviewer y8oj’s Review (Completed Before the Incident)

**Reviewer y8oj’s initial review was incorrectly attached to our submission.** This issue was corrected before the system shutdown, and the updated review—now visible in the discussion thread and revision history—was the basis for our rebuttal. We kindly ask that the final decision reflect the corrected evaluation rather than the mistaken initial review.

---

### [2] Reviewer 9QpT’s Updated Rating (Raised Before the Incident)

As recorded in the discussion thread, we provided detailed responses and additional experiments addressing the reviewer’s concerns. Reviewer 9QpT acknowledged that our rebuttal resolved the raised issues and updated the rating from 4 to 6. This update was made before the unfortunate incident, and we kindly ask that this be taken into consideration during the final decision process.

---

### [3] Summary of Our Responses to All Reviewer Comments

In our rebuttal, we addressed all reviewers’ concerns through additional experiments, clarification, and manuscript updates, as follows:

(1) We added analyses on error propagation and computational cost as requested by y8oj, clarified where diverse in-the-wild annotations can be found, and updated figures accordingly.

(2) For reviewer 7swF, we explained our approach relative to reconstruction-based methods, tracklet management, SAM2-based pipelines, and single-view template use, and added the requested visualization (Fig. 18).

(3) For reviewer 9QpT, we addressed concerns about PCA-based orientation, depth flickering, and the role of GOM, and provided new experiments on latency and depth noise; this led the reviewer to increase the score from 4 to 6.

(4) For reviewer pYBp, we clarified dataset evaluation details and configuration settings, pointing to the corresponding sections in the manuscript.

---

We made every effort to address all comments thoroughly before the incident occurred. We understand the substantial responsibilities placed on the ACs under these unusual circumstances, and we respectfully request that our responses and pre-incident reviewer updates be fully considered in the final decision.

Once again, we extend our sincere gratitude to all reviewers, the ACs, and the committee for their time and efforts during this difficult situation. We are fully happy to address any further questions or clarifications.

---

### Meta-Review · Area_Chair_NRHw · 2025-12-29

**Summary:**

This paper proposes a framework for generating 3D bounding box annotations from monocular videos without requiring ground-truth 3D supervision or LiDAR data. The method integrates off-the-shelf pre-trained models, like 2D detectors, dense point trackers, and metric depth estimators, to track objects and estimate camera motion. It introduces a Global Object Memory module to maintain identity consistency across occlusions and utilises trajectory-guided shape fusion to build pseudo-LiDAR representations for estimating 3D attributes. While the method achieves competitive results on benchmarks such as KITTI and Waymo, the technical contribution and the comparisons provided are not sufficiently convincing to warrant acceptance. For example, the framework acts primarily as an engineering assembly of existing, highly capable foundation models (GSAM, AllTracker, UniDepth). As observed by Reviewer `7swF`, the mechanisms for tracklet management and association follow standard practices in object tracking, making it difficult to pinpoint a distinct theoretical advance. Furthermore, the reliance on heuristics, such as using PCA for estimating the orientation of static objects, suggests that the geometric reasoning is not fully robust. It appears that the performance gains are largely attributable to the strength of the underlying pre-trained depth and tracking models rather than the proposed aggregation pipeline itself. Therefore, the work feels more like a successful application of recent tools rather than a significant methodological step forward for the community. This is a borderline paper. AC finds that the paper would be improved by adding more advanced methods to its framework. Thus, the paper cannot be accepted so far.

**Reviewer Concerns:**

The authors have answered specific technical questions during the discussion. For example, they provided the missing data on how long the method takes to run and how errors in depth or tracking might spread through the system, satisfying Reviewers `y8oj` and `9QpT`. They also clarified for Reviewer `pYBp` that they tested on more than just the KITTI dataset to prove the method works broadly. However, the main problem noted by Reviewer `7swF` regarding the method being a standard combination of existing tools rather than a new invention was not fully resolved. The reliance on simple rules like PCA for static objects, which Reviewer `9QpT` worried about, remains a weak point that limits how robust the system is in complex real-world scenes.

**Reviewer Scores:**

Reviewer `y8oj` corrected the initial mistake and gave a positive score of 6 after seeing the new error analysis. Reviewer `9QpT` also agreed to raise the score because the authors showed the system handles noisy depth data. Reviewer `7swF` and Reviewer `pYBp` kept their scores at 6, as they liked the results on the Waymo benchmark. Despite these borderline accept ratings, the concern reflects a competent engineering effort, difficult to make it accept at this time.

---

### Decision · Program_Chairs · 2026-01-26

Reject